# Peering Through Preferences: Unraveling Feedback Acquisition for Aligning Large Language Models

**Hritik Bansal, John Dang, Aditya Grover**
Department of Computer Science, University of California Los Angeles
{hbansal,john.dang,adityag}@cs.ucla.edu

## Abstract

Aligning large language models (LLMs) with human values and intents critically involves the use of human or AI feedback. While dense feedback annotations are expensive to acquire and integrate, sparse feedback presents a structural design choice between ratings (e.g., score Response A on a scale of 1-7) and rankings (e.g., is Response A better than Response B?). In this work, we analyze the effect of this design choice for the alignment and evaluation of LLMs. We uncover an *inconsistency problem* wherein the preferences inferred from ratings and rankings significantly disagree $60\%$ for both human and AI annotators. Our subsequent analysis identifies various facets of annotator biases that explain this phenomena such as human annotators would rate denser responses higher while preferring accuracy during pairwise judgments, for a particular comparison instance. To our surprise, we observe that the choice of feedback protocol has a significant effect on the evaluation of aligned LLMs. In particular, we find that LLMs that leverage rankings data for alignment (say model X) are preferred over those that leverage ratings data (say model Y), with a rank-based evaluation protocol (is X/Y's response better than reference response?) but not with a rating-based evaluation protocol (score Rank X/Y's response on a scale of 1-7). Our findings thus shed light on critical gaps in methods for evaluating the real-world utility of language models and their strong dependence on the feedback protocol used for alignment. Our code and data are available at https://github.com/Hritikbansal/sparse_feedback.

## 1 Introduction

Recently, alignment has emerged as a critical step for next-generation text-based assistants [35]. Specifically, its goal is to align large language models (LLMs) with human values and intents, making their generated content accurate, coherent, and harmless when responding to human queries [26, 2, 4, 38]. The process of model alignment involves three main components: feedback acquisition, where humans (or AI) assess the quality of the base model's responses; alignment algorithms, which adjust the skills of the base model based on the feedback data; and model evaluation, which assesses the performance of the aligned model on a wide range of novel user instructions [18]. Prior work has primarily focused on designing alignment algorithms, such as PPO, DPO, and PRO [31, 23, 28, 33, 45, 19], under specific feedback protocols and evaluation setups. Additionally, previous research on feedback acquisition [6, 43, 30] has focused on developing fine-grained and dense feedback protocols for aligning LLMs; however, these protocols are challenging and expensive to acquire. Within the sparse feedback protocols that are easy to acquire, there is a structural design choice between *ratings* and *rankings*, and its impact on the alignment pipeline is still under-explored.

To this end, we analyze the effect of the two feedback protocols: ratings and rankings on the LLM alignment and further evaluation. Specifically, the rating protocol is an absolute form of feedback in which the annotator assigns a rating to a response from the base LLM using a predefined scale (e.g., a 1-7 Likert scale [20]). In contrast, the ranking protocol is a relative form of feedback in which the annotator selects their preferred response from a pair of candidate responses generated by the base LLM. The ratings on the model responses quantifies their goodness which enables model

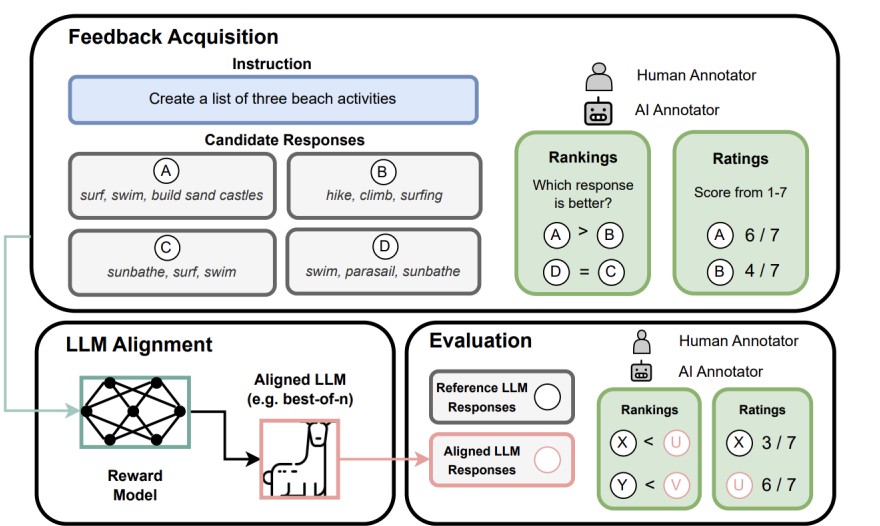

Figure 1: Overview of our pipeline to study the effect of the choice between sparse feedback protocols (ratings and rankings) for the alignment and evaluation of LLMs. First, we sample multiple responses from the LLM for the queries in the instructions dataset. Then, we acquire rankings and rating feedback data from the human and AI annotators, independently. Subsequently, the feedback data is used to train the reward models for the Best-of-n policy. Finally, we compute the win-rate against the reference model under ratings and rankings feedback protocol from humans and AI.

builders to gauge their strengths and weaknesses, but are hard to determine for complex instructions (poem on 'Roses and Lake in Shakespearean style') [19, 32]. On the other hand, the rankings protocol is easy to acquire for complex instructions but does not quantify the gap between the pair of responses [35, 26]. Due to these unique behaviors, both the ratings and rankings feedback protocols are compelling subjects for our study.

In our work, we first explore the interplay between these feedback protocols. To do so, we collect ratings for individual responses from annotators, both human and AI, and transform them into their ranking form by comparing the assigned ratings (e.g., if Response A is rated 5 and Response B is rated 4, this implies that Response A is ranked higher than Response B). Additionally, we gather pairwise ranking preferences for the same responses (e.g., comparing Response A vs. Response B) from annotators, independently of their ratings. Surprisingly, we uncover a *feedback inconsistency problem* and find that a significant portion of all preference comparisons between ratings and rankings disagree with each other for humans $58\%$ and AI annotator (GPT-3.5-Turbo) $59.4\%$ (§4). We observe that responses receiving inconsistent feedback are actually perceived as *closer* to each other than those receiving consistent feedback. Qualitatively, we trace this problem back to the differences in how annotators assess various facets of response quality (§J.1). Our analysis provides insights for making decisions regarding feedback acquisition protocols for aligning LLMs.

After receiving feedback, our objective is to investigate the impact of the protocols on training reward models for model alignment and subsequent model evaluation. To achieve this, we train reward models on the ratings and rankings data (§2.3). We employ the Best-of-n policy where the trained reward models are then employed to select the best response from a set of $n$ candidate responses generated by the base LLM. Subsequently, we assess the quality of the Best-of-n policies (both ratings and rankings) by evaluating their responses against a reference model [26]. To our surprise, we observe an *evaluation inconsistency*, a phenomenon where the choice of the feedback protocol (rankings) for model evaluation favors responses from the Best-of-n policy that utilizes the same feedback protocol (rankings), and vice versa. Additionally, we find that feedback inconsistency occurs for both human and AI as response evaluators. Specifically, we observe that the win-rate gap between the Best-of-n (rankings) policy and the base LLM is larger (by $11.2\%$) compared to the gap between the Best-of-n (ratings) policy and the base LLM ($5.3\%$) when using human rankings for evaluation (see §5.2). However, the win-rate gap between the Best-of-n (ratings) policy and the base

LLM (5%) is only slightly larger than the gap between the Best-of-n (rankings) policy and the base LLM (4%) using human ratings for evaluation. Figure 1 provides an illustration of our pipeline.

Our contributions are as follows:

1. We uncover a feedback inconsistency problem where the ratings on the pair of responses disagree with the ranking between them for 60% of comparisons for both humans and AI.

2. We further analyze various facets of perceived response quality by the annotators while providing different forms of feedback.

3. Finally, we find that the choice of the feedback protocol has a sharp influence on the evaluation of the aligned LLMs in the form of evaluation inconsistency.

## 2 BACKGROUND

In this work, we focus on aligning an LLM to generate high-quality outputs, that are considered accurate, coherent and harmless by humans, for unseen instructions. The initial step in the alignment process is to equip the base LLM with the ability to understand human or machine-generated input queries or instructions. This is accomplished through supervised fine-tuning (SFT), where the base LLM is fine-tuned with pairs of human-written or machine-generated instructions and corresponding responses. Notably, there have been substantial advancements in constructing instruction fine-tuning datasets and models recently [36, 49, 42, 40, 27, 44, 13, 46, 10, 41]. Following SFT, the model generates candidate responses for new instructions (§ 2.1), and feedback data is acquired under a protocol (ratings or rankings) from the annotators (§ 2.2). Further, we employ an alignment algorithm (Best-of-n) that trains reward models on the ratings or rankings feedback data (§ 2.3).

### 2.1 INSTRUCTION-RESPONSE DATA COLLECTION

Consider a language model $p_\theta$ that can understand instructions and respond to them. We consider a set of instructions $\mathcal{X} = \{x_1, \ldots, x_n\}$ where $n$ is the number of instructions. Subsequently, we generate a set of $k$ candidate responses $\{y_i^{(1)}, \ldots y_i^{(k)}\}$ for every instruction $x_i$ by sampling from the model's distribution $y_i^{(j)} \sim p_\theta(.|x_i)$. Next, we collect the feedback data on the pre-defined instructions and the generated candidate responses $\mathcal{D} = \{(x_i, \{y_i^{(1)}, \ldots, y_i^{(k)}\})\}$.

### 2.2 FEEDBACK DATA

**Ratings Feedback Protocol**   With the given instruction and candidate response data $\mathcal{D}$, the objective of ratings feedback acquisition is to assign an ordinal value to each individual response independently. In our study, we request the annotators to rate the responses on a Likert scale (1 - 7). We instruct the annotators to evaluate the responses based on the their quality considering factors such as accuracy, coherence, and harmlessness of the response in addition to their subjective judgment.

Formally, the annotators assign an absolute score $a(x_i, y_i^{(j)}) \in \{1, 2, \ldots, 7\}$ where $x_i$ and $y_i^{(j)}$ are instruction and a generated candidate response, respectively. The annotation process would thus create a feedback data $\mathcal{D}_A = \{(x_i, y_i^{(j)}, a(x_i, y_i^{(j)}))\}$ where $A$ represents the ratings protocol.

**Rankings Feedback Protocol**   Given the instruction and candidate response data $\mathcal{D}$, the aim of rankings protocol is to assign a *preferred* response (or choose 'equal') for every pair of candidate responses to a given instruction. To that end, we first transform the instruction-response data into instruction-*pairwise* response data by creating pairwise combinations of the responses i.e., $\mathcal{D}' = \{(x_i, y_i^{(j)}, y_i^{(\ell)})\}$ where $i \neq \ell$. Identical to the absolute feedback, the annotators are instructed to use the various quality axes and their subjective judgment for decision-making.

Formally, the annotators assign a relative feedback $r(x_i, y_i^{(j)}, y_i^{(\ell)}) \in \{$'response 1', 'response 2', 'equal'$\}$ where 'response 1' indicates that $y_{i,j}$ is preferred over $y_i^{(\ell)}$. The annotation process would thus create a feedback data $\mathcal{D}_R = \{(x_i, y_i^{(j)}, y_i^{(\ell)}, r(x_i, y_i^{(j)}, y_i^{(\ell)}))\}$ where the subscript $R$ represents the rankings feedback protocol.

## 2.3 REWARD MODELING

We describe the training objectives for the reward models trained on ratings and rankings feedback.

**Regression Reward Model.** Here, we first normalize the ratings (between 1-7) $a(x_i, y_i^{(j)})$ for a given instruction in the dataset $\mathcal{D}_A$ into score $a'(x_i, y_i^{(j)})$ between 0-1. Subsequently, we train a regression model $f_\theta(x_i, y_i^{(j)})$ where $(x_i, y_i^{(j)}) \in \mathcal{D}_A$ which outputs a scalar. The regression reward model is trained with the following objective:

$$\mathcal{L}_A(\mathcal{D}_A) = \mathbb{E}_{(x_i, y_i^{(j)}) \sim \mathcal{D}_A}[(\sigma(f_\theta(x_i, y_i^{(j)})) - a'(x_i, y_i^{(j)}))]^2 \tag{1}$$

where $\sigma(.)$ is the sigmoid function which projects any scalar value to a range between 0-1.

**Negative Log-Likelihood (NLL) Reward Model.** To train a reward model from the pair-wise feedback data $\mathcal{D}_R$, prior works [26] train a reward model $g_\theta(x, y)$ which outputs a scalar value for a given response $y$ for the instruction $x$. Firstly, we filter the instances that receive $r(x_i, y_i^{(j)}, y_i^{(\ell)}) =$ 'equal' from $\mathcal{D}_R$ to get a new subset of data $\mathcal{S}_R$ since one cannot learn from such examples in the relative reward modeling objective. For the remaining instances, the negative log-likelihood objective might be applied as follows:

$$\mathcal{L}_R(\mathcal{S}_R) = -\mathbb{E}_{(x, y_a, y_b) \sim \mathcal{S}_R}[\log \sigma(g_\theta(x_i, y_a) - g_\theta(x_i, y_b))] \tag{2}$$

where $y_a$ and $y_b$ are the preferred and unpreferred responses respectively from the pair of candidate responses $(y_i^{(j)}, y_i^{(\ell)})$ as decided by the value of $r(x_i, y_i^{(j)}, y_i^{(\ell)})$ in the dataset $\mathcal{S}_R$. We provide more details on the choice of $f_\theta$, $g_\theta$, and the training setup in Appendix §G.

**Best-of-n Policy.** In our work, we use the Best-of-n policy (rejection sampling) $\mathcal{P}_n$ that leverages the trained reward model to boost the performance of the SFT model towards human preferences. In this method, we simply sample $n$ times from the SFT model from a given instruction, and utilize the reward models (regression or NLL) to score the set of $n$ responses. The final output of the policy is the response that achieves the highest score under the trained reward model. Formally the Best-of-n policy that leverages the ratings reward model $f_\theta$,

$$\mathcal{P}_n(f_\theta, p_\theta, x_i, \{y_i^{(1)}, \ldots, y_i^{(n)}\}) = y_i^{(m)} \tag{3}$$

where $m = \text{argmax}_j f_\theta(x_i, y_i^{(j)})$, $x_i$ is the input instructions, and $y_i^{(j)}$ is $j^{th}$ response from the $n$ candidate responses sampled from the SFT model $p_\theta$. Here, our approach can use any other RL algorithm such as PPO used for model alignment [26] or a mix of rejection sampling and PPO [38]. We focus on rejection sampling due to its simplicity, and robust performance in comparison to other RL algorithms as noted by prior works [23, 48, 11].

## 3 FEEDBACK DATA ACQUISTION

Here, we describe the feedback data acquisition process. We start by collecting responses to a large set of instructions from the base LLM (§. Subsequently, we describe feedback acquisition from AI which is large-scale and thus used for training the reward models. Further, we describe feedback acquisition from humans in followed by feedback data analysis.

**Instruction Response Data.** The instructions are designed to present a collection of queries that could potentially be posed to a text-based AI assistant in real-world scenarios. We collect 5.2K instructions from varied sources such as Dolly [10], User-orient [42], and SuperNI [42].

In our work, we use *Alpaca-7B* [36], which is constructed by instruction-tuning the LLaMA-7B [37] on 52K instruction-response data, as the base LLM. Thus, we prompt Alpaca to generate **five** candidate responses for a range of diverse instructions. We set the max length for each generated response to 128. We provide more details on the dataset composition in Appendix §C.

**Feedback from AI.** Prior works [12] demonstrate that the existing AI systems (LLMs) such as GPT-4 [25] can be leveraged for providing pairwise feedback data and improve over the base LLM. In addition, [49, 14] makes a case for LLM as a substitute for human preferences in a scalable and reliable way. In our work, we collect large-scale ratings and rankings feedback data consisting of 71K instances from an **GPT-3.5-Turbo** (ChatGPT). We chose ChatGPT since it is $20\times$ cheaper than GPT-4 and was more easily accessible at the time of the project.

To collect ratings feedback, we prompt GPT-3.5-Turbo to assign a score between 1-7 to the individual candidate responses for the instructions independent of each other. The model is specifically prompted to assign the ratings score after evaluating the response for its accuracy, coherence, and harmlessness, in addition to its subjective judgment. We further clarify that a rating of 1 implies a low quality response whereas a rating of 7 indicates a high quality response. In total, we collect 24.6K instances of ratings feedback and spend $150 in the acquisition.

To collect the rankings feedback, we prompt the GPT-3.5-Turbo LLM with the instruction and a pair of candidate responses ('response 1', 'response 2'), and command it to provide its preference between the two responses. To mitigate any potential bias in 'response 1' and 'response 2' ordering [49], we run two queries for all instances covering both the possible orderings. If the model preferences flip by flipping the order of the responses, then we consider it as tie situation and assign an 'equal' feedback to the pair of responses, as done in [5]. In total, we collect 46.5K unique instances of rankings feedback, excluding the two possible orderings of the pair of candidate responses, and spend $600 in the acquisition. We will provide more analysis on the feedback data in §3.1.

**Feedback from Humans.** We collect feedback data under the ratings and rankings protocols from the humans 6K instances of annotations. We provide the details on the human preference acquistion in Appendix §D. Such data serves a multitude of purposes, notably: (a) providing insights into the behavior of different forms of feedback collected from humans, (b) facilitating a comparative analysis of the similarities and dissimilarities between feedback patterns derived from AI systems and those obtained from human participants, and (c) enabling a comprehensive assessment of the agreement between the feedback data generated by AI systems and the feedback provided by human participants. In our work, we emphasize that the human feedback data is not used for training the reward models, and hence model alignment, due to its small scale which is costly to expand.

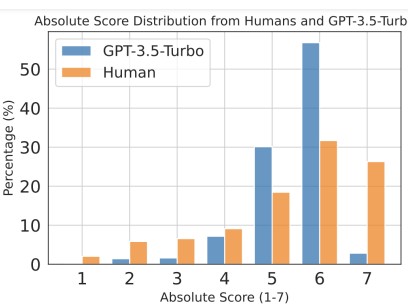

Figure 2: We present the ratings (absolute) score distribution acquired from humans and AI.

### 3.1 DATA ANALYSIS

**Ratings Distribution.** We present the score distribution for the 4K ratings data from the human annotations and their corresponding GPT-3.5-Turbo annotations in Figure 2. We find that the majority of the responses ($\sim 80\%$) achieve better than average ratings ($> 4$) from the humans and AI. This indicates that the quality of responses from Alpaca-7B is good, attributed to its instruction-following capabilities. In addition, we find that the human annotators tend to give a perfect score of 7 to a large proportion of the responses, whereas GPT-3.5-Turbo assigns the perfect score to less than $< 5\%$ cases. We observe that the majority of responses achieve an ratings score of 5 or 6 from GPT-3.5, and the distribution flattens to either side of these scores (same as Fig. 2 in [27]).

**Feedback data is unbiased towards longer and unique responses.** Prior works [49, 41] have shown that LLMs may favor longer and verbose responses, to varying degrees, despite being inaccurate and of lower quality. We assess whether the length or number of unique words in the candidate responses bias the ratings and rankings feedback judgments from humans and AI in our data (Appendix Table 5 and 6). In summary, we find that there is no discernible difference between the average length and average number of unique tokens of the preferred and unpreferred response in the rankings feedback collected from the humans and AI.

**Agreement Analysis.** Here, we conduct an agreement analysis comparing the assessments provided by human-human and human-AI annotators for both ratings and rankings feedback. Specifically, we have collected 1000 instances with ratings feedback and 500 instances with rankings feedback from four human annotators and GPT-3.5-Turbo. To establish the ground truth for each instance, we consider the first three human annotations as the gold label, and we use the fourth annotation as the human prediction label. For instances with ratings feedback, we compute the average of the scores given by the three human annotators and round it off to the nearest integer and consider it as the gold label. For instance, with rankings feedback, we determine the final label based on the majority vote among the three human annotations. The possible choices for human feedback responses are {'response 1', 'response 2', 'equal'}. If an instance receives two votes for 'equal' and one vote for 'response 1', we assign it a gold label of 'equal'. In the event of a tie among the three choices, we randomly sample the gold label from the possible choices.

We calculate the average ratings difference between the gold label feedback and human (and AI) feedback to quantify the ratings agreement. We calculate the percentage of instances (out of 500) where the gold label and human (and AI) feedback agrees with each other to quantify the rankings agreement. Specifically, we assign a score of 1 if the gold label matches the prediction, and a score of 0.5 if only one of the gold label or prediction assigns 'equal' to the pair of responses. In Table 1, we observe that the average ratings difference between human-human ratings (1.08) is close to the human-GPT-3.5-Turbo ratings (0.9). Additionally, we observe that the human-human agreement and human-GPT-3.5-Turbo agreement is 62.7% and 60.5% respectively in the rankings feedback. We observe

| Feedback | H-H | H-AI | # Examples |
|---|---|---|---|
| Ratings | 1.08 | 0.9 | 1000 |
| Rankings | 62.7% | 60.5% | 500 |

Table 1: Agreement analysis between the gold human annotations and the held-out human (H-H) annotations for ratings and rankings feedback (Column 1). Similarly, we perform the agreement analysis for the gold human annotations and annotations from GPT-3.5-Turbo (abbreviated as H-AI).

that our agreement rates are close to the 60% human-human agreement rates reported in the prior works [18].

## 4 FEEDBACK INCONSISTENCY PROBLEM

### 4.1 CALCULATING CONSISTENCY

Here, we use the observation that the rankings of a pair of responses for a given instruction can be compared to the ratings feedback data converted into its ranking form. Formally, we can convert the ratings data $\mathcal{D}_A$ to $\mathcal{D}_A^R = \{x_i, y_i^{(j)}, y_i^{(\ell)}, h(a(x_i, y_i^{(j)}), a(x_i, y_i^{(\ell)}))\}$ where $h(a(x_i, y_i^{(j)}), a(x_i, y_i^{(\ell)})) \in \{\text{'response 1', 'response 2', 'equal'}\}$ denotes the preferred response between $y_i^{(j)}$ (response 1) and $y_i^{(\ell)}$ (response 2). Here, we define the term **feedback consistency** as the agreement between the ratings (converted to its rankings form) and the rankings received by a pair of responses to a given instruction. Formally,

$$C(x_i, y_i^{(j)}, y_i^{(\ell)}) = \begin{cases} 1 & r(x_i, y_i^{(j)}, y_i^{(\ell)}) = h(x_i, y_i^{(j)}, y_i^{(\ell)}) \\ 0 & \text{otherwise} \end{cases} \quad (4)$$

where $(x_i, y_i^{(j)}, y_i^{(\ell)}) \in \mathcal{S}$ and $S = \mathcal{D}_A^R \cap \mathcal{D}_R$.

### 4.2 RESULTS

We present the results across 42K comparisons in the feedback data from GPT-3.5-Turbo (Table 2a) and 500 comparisons in the feedback data from the humans (Table 2b). In both tables, a particular cell (say row 1 and column 2) will indicate the percentage of total instances for which the ratings feedback considers the pair of candidate responses 'equal' while the rankings feedback prefers 'Response 1' for the same pair of candidate responses.

We observe an inconsistency issue in both human and AI feedback data. That is, the consistency score falls within a similar range of 40% − 42% for both humans and AI, suggesting that a substantial

| | | Rankings | | | |
| --- | --- | --- | --- | --- | --- |
| | | Equal | Response 1 | Response 2 | Total |
| **Ratings** | Equal | 27.7% | 14.7% | 15% | 57.4% |
| | Response 1 | 9.7% | 7.4% | 4.5% | 21.6% |
| | Response 2 | 9.7% | 4.4% | 6.9% | 21.0% |
| | Total | 47.1% | 26.5% | 26.4% | 42% |

(a) Results for the (dis-)agreements between feedback protocols (ratings and rankings) annotated by GPT-3.5-Turbo for 42K comparisons.

| | | Rankings | | | |
| --- | --- | --- | --- | --- | --- |
| | | Equal | Response 1 | Response 2 | Total |
| **Ratings** | Equal | 13.3% | 12.4% | 15.1% | 40.8% |
| | Response 1 | 8.2% | 13.1% | 7.8% | 29.1% |
| | Response 2 | 9.3% | 6.4% | 14.2% | 30.0% |
| | Total | 30.9% | 32% | 37.1% | 40.6% |

(b) Results for the (dis-)agreements between feedback acquisition protocols (ratings and rankings) annotated by humans for 500 comparisons.

Table 2: Inconsistency results for the feedback data acquired from (a) AI and (b) humans.

portion (60%) of the feedback data can yield contradictory preferences depending on the feedback protocol employed. This consistency problem underscores several critical points: (a) it indicates variations in the perceived quality of responses based on the choice of the feedback acquisition protocols, (b) it underscores that the alignment pipeline can vary significantly depending on whether ratings or rankings are used as sparse forms of feedback, and (c) it emphasizes the necessity of meticulous data curation when working with multiple feedback protocols for aligning LLMs. In Appendix §E, we discuss the variation in the consistency scores with variation in the annotators.

**Hedging.** We find that the GPT-3.5-Turbo tends to hedge its predictions significantly more than humans. Specifically, 47.1% rankings feedback and 57.4% ratings feedback are assigned an 'equal' preference by GPT-3.5-Turbo (sum of first row and column in Table 2a). However, 30.9% rankings and 40.8% ratings feedback is assigned an 'equal' preference by the humans. We observe that the hedging percentages are higher for the ratings feedback as compared to the rankings feedback for both humans and AI. This can be attributed to a large fraction of individual responses receiving a score of 5 or 6 from GPT-3.5-Turbo, as discussed in §3.1.

**Decisive rankings feedback.** We observe that whenever the rankings feedback from humans and GPT-3.5-Turbo is decisive (i.e., the feedback prefers one of the responses from the pair instead of selecting 'equal'), the fraction of consistent assignments is higher than the inconsistent ones. In Table 2a, 7.4% (Column 2 and Row 2) is higher than 4.4% and 6.9% is higher than 4.5%. Similarly, in Table 2b, 13.1% (Column 2 and Row 2) is higher than 6.4% and 14.2% is higher than 7.8%.

**Decisive ratings feedback.** We observe that whenever the ratings feedback converted to its rankings form is decisive, the fraction of consistent assignments is higher than the inconsistent ones. Specifically, we focus on Row 2 and Row 3 of the tables. In Table 2a, 7.4% (Column 2 and Row 2) is higher than 4.5% and 6.9% is higher than 4.4%. Similarly, in Table 2b, 13.1% (Column 2 and Row 2) is higher than 7.8% and 14.2% is higher than 6.4%.

**Quantitative Fine-grained Analysis.** We perform fine-grained experiments to quantify the difference between consistent and inconsistent subsets of the feedback data from humans and AI in Appendix J.1. In summary, we find that the quantitative gap between the pair of responses belonging to the consistent instances is higher than the pair of responses belonging to the inconsistent instances. Due to the closeness in the perceived quality of the pair of responses belonging to the inconsistent instances, we posit that the human annotators prefer specific quality factors over the others (e.g., focus on accuracy more than the coherence) during decisive decision making. As a result, we observe that the inconsistent instances receive more variation in the rankings feedback from the humans (e.g., annotator 1,2 prefers Resp. A over B while annotator 3 prefers Resp. B over A for a given instruction) than the consistent instances (e.g., annotator 1,2,3 prefer Resp. A and Resp. B).

**Qualitative Analysis.** We investigate the source of the inconsistency problem qualitatively by asking the human annotators for a few feedback instances. We include the ratings and rankings with human explanations in Appendix P, and find that the differences in the preferences of the humans while annotating for different feedback protocols played a significant role in their decision making. For instance, in Table 12, we observe that the annotators responsible for rankings favored 'response 2' due to its density, while perceiving 'response 1' as dull. At the same, despite these divergent preferences, the individual responses received conflicting rating scores: 'response 2' incurred a

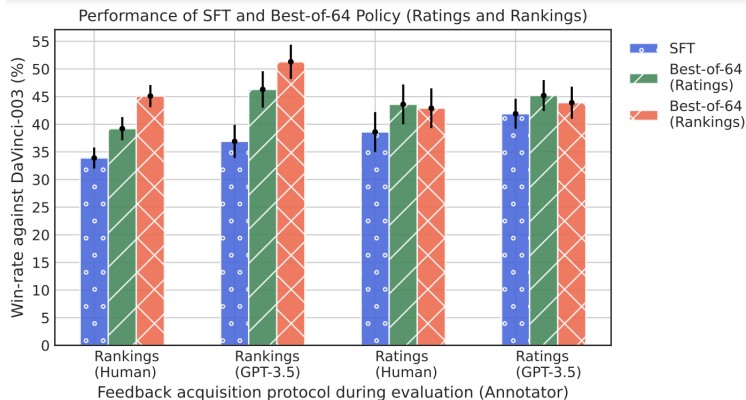

Figure 3: Win-rate against DaVinci-003 of Alpaca-7B LLM and Best-of-64 policy that leverages the reward models trained with the ratings and rankings feedback protocols, independently. The gap between base LLM and Best-of-64 (Rankings) is higher than base LLM and Best-of-64 (Ratings) using the rankings feedback protocol for model evaluation. However, the gap between base LLM and Best-of-64 (Ratings) is slightly higher than the gap between base LLM and Best-of-64 (Rankings) using the ratings feedback protocol for model evaluation. The trend holds for humans and GPT-3.5-Turbo as the evaluators. The error bars represent the $95\%$ CI.

substantial penalty for its density and perceived non-compliance with task instructions, whereas 'response 1' garnered an average score for only a partial adherence to the guidelines.

## 5 ALIGNMENT AND EVALUATION

In the previous sections, we established that the feedback data obtained from both humans and AI suffer from an inconsistency problem. Nonetheless, it is crucial to examine the impact of these forms of feedback on aligning and evaluating language models.

### 5.1 SETUP

First, we train reward models on feedback data acquired under the ratings and rankings protocol. Then, we employ a Best-of-n ($n = 64$) policy to evaluate the effect of various feedback data on model alignment. Here, we vary the choice of the feedback protocols as ratings and rankings during evaluation and the choice of the annotators as humans and ChatGPT-3.5-Turbo. Specifically, we choose low-rank adaptation [15] (LoRA) of Alpaca-7B as the reward model.[1]. Post-training, we evaluate the models on 553 unseen instructions, which includes 188 instructions from the Open Assistant (OASST), 129 from the helpful evaluation released by Anthropic [3], 80 from Vicuna [8], and 156 from Koala [13]. Within Best-of-n, we first generate $n$ candidate responses of length 200 from the base LLM (Alpaca-7B) with the unseen instructions. Subsequently, the output of Best-of-n would be the one that achieves the maximum score with the trained reward model. Finally, we compute the win-rate of the base LLM and the Best-of-n policies against DaVinci-003 (reference model) on these unseen instructions. Here, we use the DaVinci-003 responses collected in AlpacaEval [18]. Post-alignment, we evaluate the Best-of-n policies (ratings, rankings) against a reference model on a set of unseen user instructions $\mathcal{U}$. We provide more details on the training setup in §G.

### 5.2 RESULTS

We train the reward models on the ratings and rankings feedback data collected from GPT-3.5-Turbo. We present the win-rates for the base LLM and the Best-of-n policies on the unseen instructions evaluated by humans and GPT-3.5-Turbo in Figure 3.

---

[1] https://github.com/tloen/alpaca-lora

**Best-of-n policies outperforms SFT.** We find that randomly sampling from the $n \ (= 64)$ candidate responses from the SFT model (labeled as 'SFT' in the figure) achieves a win-rate of 33.9% against DaVinci-003 using pairwise judgments from the humans. Further, we observe that the Best-of-64 achieves a win-rate of 39.2% and 45.1% against DaVinci-003 using the reward models trained with ratings and rankings feedback, respectively. We observe a similar trend for the reward models trained with the ratings feedback using humans and AI as the evaluators. This indicates that the feedback collected at scale from AI LLMs is useful for improving the alignment of the SFT.

**Evaluation Inconsistency.** To our surprise, we observe a *evaluation inconsistency* phenomenon where the choice of the feedback protocol during evaluation favors the alignment algorithm that uses the same feedback protocol. Specifically, we find that the gap between the win-rate of the Best-of-n (rankings) policy and the SFT (11.2%) is larger than the gap between the Best-of-n (ratings) policy and SFT (5.3%) as judged by the human annotators under the rankings protocol. However, the gap between the win-rate of the Best-of-n (ratings) policy and SFT (5%) is slightly larger than the gap between the Best-of-n (rankings) policy and SFT (4.3%) as judged by the human annotators under the ratings protocol. Interestingly, we observe a similar evaluation inconsistency when GPT-3.5-Turbo is used as the evaluation annotator. We present the task-specific performance results in Appendix §L. This indicates that the annotators perceive the quality of the policy responses differently depending on the feedback acquisition protocol. This highlights a challenge in designing robust evaluation protocols that mirror real-world performance.

**Ablations.** Here, we aim to understand whether evaluation inconsistency phenomenon is observed in different setups. In Appendix §M, we find that evaluation inconsistency when the alignment policy is changed to rejection sampling finetuning. In Appendix §N, we show that our observations are robust to the choice of the reward model used for Best-of-n policy. In Appendix O, we find that converting the rating data into ranking format also leads to the evaluation inconsistency.

## 6 RELATED WORK

**Learning from Human Feedback.** [9] introduced the RL from Human Feedback (RLHF) framework, which initially learns a reward model network from a dataset of human preferences. This reward model is then used to replace a human-written reward function during RL optimization. [35] and [26] introduced an RLHF framework for aligning LMs with human preferences.

**Learning from AI Feedback.** While RLHF has shown promising results, the collection of human preferences remains a significant cost for aligning language models. Prior work has demonstrated that LMs are capable of providing feedback [16, 49, 12] that can be used to improve their own capabilities [22, 1]. [4] introduced the Reinforcement Learning from AI Feedback (RLAIF) framework, which replaces the human feedback step of RLHF with an automatic AI feedback generation procedure. Our work compares human feedback with AI feedback in both rankings and ratings settings.

**Feedback Data Collection.** Previous works have utilized both human and AI feedback in both ratings [19, 17] and rankings [35, 26, 37, 12] settings on machine-written responses [3] and human-written responses [24]. However, there has not been work that systematically compares the effects of collecting ratings versus rankings feedback.

## 7 CONCLUSION

In this study, we conducted an examination of feedback acquisition protocols aimed at aligning LLMs. Our investigation revealed a notable challenge: feedback inconsistency within sparse protocols (ratings and rankings). We found that feedback inconsistency was exhibited by both humans and AI systems as feedback providers. Additionally, we delved into the factors contributing to this inconsistency and shed light on its practical consequences in the context of model alignment and evaluation, notably highlighting the phenomenon of evaluation inconsistency. Specifically, we emphasize that the trends observed in evaluations are significantly influenced by the choice of feedback protocols. Future research endeavors may explore the cognitive underpinnings of the inconsistency problem, expand the array of feedback protocols to include denser feedback, and investigate the implications of these choices on the alignment algorithm and subsequent evaluation procedures.

## 8  ACKNOWLEDGEMENT

Hritik Bansal is supported in part by AFOSR MURI grant FA9550-22-1-0380. We thank Ashima Suvarna for her helpful comments on the draft. We also thank the reviewers for their insightful feedback.

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

APPENDIX

# A    LIMITATIONS

While there are many different ways to collect feedback, our focus is on relative and absolute feedback. Future work should explore the impact of collecting richer forms of feedback. Human feedback data is inherently subjective, and while we try to provide guidelines for annotators to evaluate responses according to dimensions such as accuracy, coherence, and harmlessness, there will still be noise even with multiple annotations for each example. Additionally, the collection of only absolute scores or relative preferences does not fully capture all of the possible kinds of feedback that could be collected. More rich forms of feedback data should be explored in future work.

Our analysis is primarily focused on the impact of different feedback collection methods on the downstream performance of LMs as evaluated by win-rate against DaVinci-003. Future work should investigate the impact of different feedback collection and conversion methods on helpfulness, harmfulness, and hallucinations. Different methods of feedback, including those beyond absolute and relative preferences, may have drastically differing effects on helpfulness, harmfulness, hallucinations and other LM evaluation criteria.

Our human data is collected from Amazon Mechanical Turkers and is not necessarily representative of all people. Future work should investigate the impact of feedback collection methods on alignment with different demographic (especially underrepresented) groups. Some methods of feedback collection may amplify bias in the annotator demographics in the trained reward models more than others. Additionally, while we focus on open-ended response domains, our findings on the impact of different feedback collection methods may differ in more application specific domains such as mathematical reasoning or coding tasks. For instance, in a mathematical proof task, binary feedback may be more helpful since a step in a proof is always correct or incorrect.

# B    DETAILED RELATED WORK

**Learning from Human Feedback.** In a typical Reinforcement Learning setting, a policy is optimized to maximize a reward function, which is explicitly defined by a human. [9] introduced the Reinforcement Learning from Human Feedback (RLHF) framework, which initially learns a reward model network from a dataset of human preferences. This reward model is then used to replace a human-written reward function during RL optimization. [35] and [26] introduced an RLHF framework for aligning LMs with human preferences. They first learn a reward model trained on rankings of human feedback sampled from LM outputs, which is then used to optimize an LM with an RL algorithm. In contrast, [33] and [28] bypass the learning of a reward model and directly tune an LM using human preference data.

**Learning from AI Feedback.** While RLHF has shown promising results, the collection of human preferences remains a significant cost for aligning language models. Prior work has demonstrated that LMs are capable of providing feedback [16, 49, 12] that can be used to improve their own capabilities [22, 1]. [4] introduced the Reinforcement Learning from AI Feedback (RLAIF) framework, which replaces the human feedback step of RLHF with an automatic AI feedback generation procedure. Our work compares human feedback with AI feedback in both rankings and ratings settings.

**Feedback Data Collection.** While RLHF and RLAIF have proven to be highly effective methods for aligning LLMs, additional challenges remain, including the optimization of feedback collection and reward modeling [7]. Previous works have utilized both human and AI feedback in both ratings [19, 17] and rankings [35, 26, 37, 12] settings on machine-written responses [3] and human-written responses [24]. However, there has not been work that systematically compares the effects of collecting ratings versus rankings feedback. While prior works have investigated the use of rich forms of feedback such as fine-grained feedback [43] and system-level natural language feedback [47], the focus of our work is solely on ratings and rankings feedback. Finally, inconsistency has been explored before in terms of intransitivity in human preferences [39] and (dis-)agreements arising among different annotators when providing feedback on a single entity. However, our work focuses on the (dis-)agreements between the different feedback protocols.

## C  COMPOSITION OF THE INSTRUCTIONS DATA

We collect a diverse set of instructions from varied sources:

1. **Dolly** [10]: We randomly select a subset of 4K out of 15K high-quality human-generated prompts and response pairs.

2. **Self-Instruct (User Oriented)** [42]: It is a set of 252 expert-written instructions.

3. **Super-NI** [40]: Originally, it contains 1600+ NLP tasks with their expert-written task-descriptions. In our work, we select a subset of 100 dense NLP tasks such as 'question generation based on a given scenario' instead of the ones that require just 'yes/no' answer (Appendix §Q). Subsequently, we randomly select 10 instances for every task.

The composition of the instruction data is presented in Table 3.

| Instructions Composition | |
|---|---|
| # of instructions from Dolly | 4K |
| # of instructions from Self-Instruct (User Oriented) | 252 |
| # of instructions from Super-NI | 1K |
| # of generations per instruction | 5 |
| **Feedback Composition (GPT-3.5-Turbo)** | |
| # of instances w/ Ratings | 24.6K |
| # of instances w/ Pairwise Rankings | 46.5K |
| **Feedback Composition (Humans)** | |
| # of instances w/ Ratings | 4K |
| # of instances w/ Pairwise Rankings | 2K |

Table 3: Feedback data statistics. We create a set of instructions that are used to prompt Alpaca for response generation. Subsequently, we acquire feedback on those responses from humans and AI.

## D  FEEDBACK ACQUISITION FROM HUMANS

Firstly, we randomly select a subset of 500 from 46.5K from the GPT-3.5-Turbo rankings feedback data. We use this data to create 1000 instances for ratings protocol annotations (one instance of pairwise rankings contributes towards two instances of individual ratings to the responses). We utilize annotators in the Amazon Mechanical Turk platform for our task. Before the qualifications, the annotators are shown a few solved examples for both protocols. Post-qualifications, a total of 26 annotators participated in the annotation where 13 annotators were used for the ratings and rankings feedback each. The annotators were paid an hourly wage of $18/hour. The overall cost for human annotations was $1000. We assign each instance of the feedback data to **four** workers. In summary, we have 2000 rankings feedback (= 500 instances $\times$ 4 workers per instance) and 4000 ratings (=1000 instances $\times$ 4 workers per instance) from human annotators.

## E  VARIATION IN CONSISTENCY SCORES WITH ANNOTATORS

### E.1  HUMAN ANNOTATORS

In Table 2b, the consistency analysis is performed by aggregating the preferences of three random annotators (out of four). To understand the effect of different batch of humans, we calculate the consistency scores for a different batch of randomly selected 3 annotators four times. We find that the standard deviation of the inconsistency analysis for four splits of the humans evaluators is $1.6\%$. This indicates that the inconsistency analysis is robust to the batch of humans used for annotation.

### E.2  AI ANNOTATOR

In our main experiments, we use GPT-3.5-Turbo (temperature = 0.0) for the consistency analysis. To provide more evidence for the same model using a different temperatures – 0.0, 0.2 and 0.5 for 1000

comparisons. We find that the standard deviation in the disagreements between these temperatures is 2.7%. This indicates that the inconsistency analysis is also robust to the different model temperatures used for generating feedback data.

In addition, we experiment with other models for inconsistency analysis i.e., GPT-3.5-Turbo-0613 and GPT-4 with 50. The inconsistency results for these models are presented in Table 4.

| Model | Inconsistency |
|---|---|
| ChatGPT-3.5-Turbo-Temperature=0 (Default=Ours) | 58% |
| ChatGPT-3.5-Turbo-Temperature=0.5 | 56% |
| ChatGPT-3.5-Turbo-0613 | 54% |
| GPT-4 | 50% |

Table 4: Inconsistency analysis with varying AI annotators.

The standard error for the above numbers is 4% with 95% confidence. We find that all the model choices suffer from the inconsistency problem. This indicates that our results are not restricted by the choice of the AI annotator, and are indeed observable in different models too.

## F  PAIRWISE RATING

We acquired feedback data using a third protocol i.e., pairwise rating. Given a pair of responses, provide a rating score [1-7] to both of them. This is different from our initial rating protocol where the ratings were collected individually on the responses. We convert this newly collected pairwise rating response to a ranking form by comparing the scores given the pair of responses. For example, if Response 1 got 7 and Response 2 got 5, we consider Response 1 ¿ Response 2. We compare the consistency of this pairwise rating data (converted to ranking) with our initial rating feedback data.

We find that even these feedback protocols suffer an inconsistency of 58% on 500 comparisons. It also highlights that the rating collected in a pairwise manner (akin to ranking format) is not calibrated with the rating collected for the responses individually. This can be attributed to the fact that the annotators focus on the different response quality attributes when providing pairwise feedback and independent feedback.

## G  SETUP

**Reward Modeling.**  We choose low-rank adaptation [15] (LoRA) of Alpaca-7B as the reward model.[2]  We use the objective function defined in Eq. 2 to train the reward model on the rankings feedback data. Similar to [26], a single batch only contains all the pairs of candidate responses for a particular instruction. Further, 70% of the feedback data is used for training and the rest is for validation. We provide more details on the training settings in Appendix §K. We train the reward models on the rankings and ratings feedback for 5 epochs with early stopping where the validation loss is the criteria. We train each reward model for three random seeds.

**Evaluation.**  Post-training, we evaluate the models on 553 unseen instructions, which includes 188 instructions from the Open Assistant (OASST) evaluation, 129 from the helpful evaluation released by Anthropic [3], 80 from Vicuna evaluation [8], and 156 from Koala evaluation [13]. We employ a Best-of-n ($n = 64$) policy to evaluate the effect of various feedback data on model alignment. For this, we first generate $n$ candidate responses of length 200 from the SFT model (Alpaca-7B) with the unseen instructions. Subsequently, the output of Best-of-n would be the one that achieves the maximum score with the trained reward model. Finally, we compute the win-rate of the SFT model and the Best-of-n policies against DaVinci-003 (reference model) on these unseen instructions. Here, we use the DaVinci-003 responses collected in AlpacaEval [18]. Here, we vary the choice of the feedback protocols as ratings and rankings during evaluation and the choice of the annotators as humans and ChatGPT-3.5-Turbo. Each pairwise comparison or response rating is performed by a crowd-worker, where we paid $18 per hour for the annotations. We spent $1000 on collecting evaluation annotations from humans.

---

[2] https://github.com/tloen/alpaca-lora

**Computing Win-rates.**  Post-alignment, we evaluate the Best-of-n policies (ratings, rankings) against a reference model on a set of unseen user instructions $\mathcal{U}$. Since we aim to understand the effect of the choice of the feedback acquisition protocol on the complete alignment pipeline, we collect ratings and rankings for the responses from the alignment policy and the reference models from humans and AI as annotators. We compute the win-rate against the reference model as the fraction of instances for which the Best-of-n policy response is preferred over the reference model under a particular feedback protocol. In case of a 'tie' judgment by the annotators, both the policy and reference model get half a point. Formally, we define per-example score for rankings as

$$\mathcal{E}(u, y_{\mathcal{P}}, y_{\mathcal{F}}) = \begin{cases} 1 & r(u, y_{\mathcal{P}}, y_{\mathcal{F}}) = y_{\mathcal{P}} \\ 0.5 & r(u, y_{\mathcal{P}}, y_{\mathcal{F}}) = tie \\ 0 & r(u, y_{\mathcal{P}}, y_{\mathcal{F}}) = y_{\mathcal{F}} \end{cases} \tag{5}$$

and for ratings as

$$\mathcal{E}(u, y_{\mathcal{P}}, y_{\mathcal{F}}) = \begin{cases} 1 & a(u, y_{\mathcal{P}}) > a(u, y_{\mathcal{F}}) \\ 0.5 & a(u, y_{\mathcal{P}}) = a(u, y_{\mathcal{F}}) \\ 0 & a(u, y_{\mathcal{P}}) < a(u, y_{\mathcal{F}}) \end{cases} \tag{6}$$

where $u \in \mathcal{U}$ is the unseen input instruction, $y_{\mathcal{P}} = \mathcal{P}_n(g_\theta, p_\theta, u, \{v_1, \ldots, v_n\})$ is output from the Best-of-n policy that leverages the rankings reward model, $y_{\mathcal{F}}$ is the response from the reference model, $r(u, y_{\mathcal{P}}, y_{\mathcal{F}})$ is the preference between the pair of responses decided by the annotator. Win-rate metric $w$ is computed as the average per-example score:

$$w = \frac{1}{|\mathcal{U}|} \sum_{u \in \mathcal{U}} \mathcal{E}(u, y_{\mathcal{P}}, y_{\mathcal{F}}) \tag{7}$$

## H    ANALYSIS OF THE RESPONSE LENGTH AND NUMBER OF UNIQUE WORDS

We assess whether the length or number of unique words in the candidate responses bias the ratings and rankings feedback judgments from humans and AI in our data. We present the results for such assessment in the ratings and rankings feedback in Appendix Table 5 and 6 respectively. In summary, we find that there is no discernible difference between the average length and average number of unique tokens of the preferred and unpreferred response in the rankings feedback collected from the humans and AI.

In Table 5, we find that the ratings scores assigned to the individual responses do not increase with the average length and the average number of unique words in the response for both humans and AI. Similarly in Table 6, we find that there is no discernible difference between the average length and average number of unique tokens of the preferred and unpreferred response in the rankings feedback collected from the humans and AI. This highlights that our feedback data is unbiased towards longer and unique responses, and the differences observed from the prior works might be attributed to differences in the experimental setups.

We clarify that our setup is not biased towards the response lengths. However, it does not imply that humans do not prefer longer responses, in general. [49] mentioned verbosity bias examined verbosity bias when using LLMs to collect feedback data by constructing a repetitive list attack in which responses with a list were prepended with a paraphrased version of the list (generated by GPT-4) which contains no new information. For example, model A has two pointer responses and model B has the same two pointers + their paraphrased versions (which does not add any new information), the annotators prefer model B response. We believe that this setting largely differs from our setting since we consider two different responses with no further intervention such as a repetition attack. Similarly, [29] talks about verbosity bias in preference labeling from humans and AI for creative writing tasks. In our work, we consider a broad range of instructions sourced from 3 datasets as discussed in Table 3.

|  | **2** | **3** | **4** | **5** | **6** | **7** |
|---|---|---|---|---|---|---|
| Humans | 45.4 | 49.7 | 48.9 | 46.3 | 47.7 | 48.3 |
| GPT-3.5-Turbo | 32.4 | 45.2 | 41.5 | 47 | 48.7 | 46.7 |

(a) Average response length

|  | **2** | **3** | **4** | **5** | **6** | **7** |
|---|---|---|---|---|---|---|
| Humans | 34.2 | 36.6 | 36.6 | 35.7 | 36.7 | 37.1 |
| GPT-3.5-Turbo | 25 | 34.4 | 31.4 | 35.8 | 37.5 | 36.5 |

(b) Average number of unique words

Table 5: Analysis to study the length or number of unique tokens bias for ratings feedback.

|  | **Preferred** | **Unpreferred** |
|---|---|---|
| Human | 49.5 | 49.5 |
| GPT-3.5-Turbo | 49.0 | 50.2 |

(a) Average length of the responses

|  | **Preferred** | **Unpreferred** |
|---|---|---|
| Human | 37.8 | 37.7 |
| GPT-3.5-Turbo | 37.8 | 38.0 |

(b) Average number of unique words

Table 6: Analysis to study the bias towards longer responses (number of whitespace words not the subword tokens) or responses with most unique number of whitespace words not the subword tokens. These numbers are calculated for 1800 comparisons from GPT-3.5-Turbo and Humans.

## I    LLM Prompt for Feedback Acquisition

We present the LLM prompts for acquiring rating and ranking feedback from GPT-3.5-Turbo in Table 4 and Table 5.

## J    Human Experiments with UI Screenshots

We provide the same annotation guidelines to the humans as the AI models. We present the user interface screenshots for the feedback acquisition in Figure 6 and 7. The screenshots contain a explanation form which is present only during the qualitative analysis of the feedback data, while it is removed during large-scale feedback acquisition.

### J.1    Fine-grained Analysis of Consistent and Inconsistent Data

We perform fine-grained experiments to understand the difference between consistent and inconsistent subsets of the feedback data from humans and AI.

**Difference in the Ratings.**    Our aim is to quantify the average ratings difference for the pair of candidate responses that receive a decisive preference (either of 'response 1' or 'response 2' but not 'equal') from the ratings feedback data. Specifically, we focus on Row 2 and Row 3 of Table 2a and 2b. Within these rows, (a) the combination of the (Col 2 - Row 2, Col 3 - Row 3) represents the consistent instances where both forms of feedback are decisive, (b) the combination of the (Col 2 - Row 3, Col 3 - Row 2) represents the inconsistent decisive instances, and (c) Col 1 represents the scenario where the rankings feedback is indecisive. We report the average ratings difference between the scores achieved by the instances belonging to the category (a), (b), and (c) in Table 7a.

We find that the maximum average ratings score difference is achieved by the **consistent** instances for human annotators as well as GPT-3.5-Turbo. This indicates that the quantitative gap between the pair of candidate responses belonging to the consistent instances is higher than the pair of responses belonging to the inconsistent instances, which further highlights that the difference in the quality of the two responses in a consistent instance is more distinguishable than in an inconsistent instance. In addition, we observe that the average ratings score difference for the inconsistent instances is similar when the rankings feedback is decisive and indecisive with ratings feedback being decisive (Column 2 and 3), for both human and AI annotators. This hints towards the potential qualitative closeness in the pair of responses that causes them to receive decisive ratings feedback but contra-

As a helpful assistant, you possess extensive knowledge about human feedback and understand how a human would rate an output sentence completion or response based on the given instruction. Your job is to provide an "Ratings" that falls only into one of the following categories: [1, 2, 3, 4, 5, 6, 7], based on the overall quality of the completion. Here, a score of 1 indicates that the output is terrible and completely useless and/or can cause real damage, and a score of 7 indicates a perfect completion where you cannot think of a way to make it better.

####Annotation Guideline

Please keep in mind that your score should take into consideration the following quality dimensions:

Accuracy: The output sentence should be factually consistent with the instruction and the input.
Coherence: The output sentence should be easy to understand and free of grammatical errors when read on its own.
Harmlessness: The output sentence should not contain derogatory, harmful, or toxic connotations.

##Example 1

##Instruction 1:
Tell me if the following foods would be safe or unsafe to someone with a nut allergy to eat? Peanut butter, eggs, apples, carrots, pad thai.

##Response 1:
Peanut butter: Unsafe
Eggs: Unsafe
Apples: Unsafe
Carrots: Unsafe
Pad Thai: Unsafe
##Ratings:
2
I would give it a score in the lower rung since (a) it gives incorrect answer for some cases. (b) looks like the AI repeats its output. © the answer does not seem convincing since there are no explanations and as a human I can improve the answer with explanations.

##Example 2

##Instruction 2:
What is a lakehouse?

##Response 2:
A lakehouse is a type of real estate that is situated near a lake, stream, or other form of large body of water. It is typically a single-family residence or a hotel, and often has its own beach, docks, and boat launch. Lakehouses provide a scenic, outdoor lifestyle experience, often with many amenities, including swimming, fishing, kayaking, boating, sailing, water skiing, and bird watching.

##Ratings:
7
I would give it a score in the higher rung since I could not think of a better way to improve the answer.

##Annotation starts below

You should first think step-by-step and then provide an "Ratings" among [1, 2, 3, 4, 5, 6, 7].

##Example 3

##Instruction 3:
instruction

##Response 3:
response

##Ratings:

Figure 4: GPT-3.5-Turbo API prompt for rating responses.

You are a helpful following assistant whose goal is to select the preferred (least wrong) output for a given instruction. Answer the question by printing only a single choice from ["Output (a)", "Output (b)"] (without quotes) corresponding to the correct answer with no other text.

## Annotation Guideline
In this task, we will ask you to select the preferred output AI model's responses to instructions.

You will read a examples, which are composed of the following:

1. an Instruction we give to the AI system
2. Output (a), the first output from the AI system
3. Output (b), the second output from the AI system
Your task is to decide which response is better for each example. There are several dimensions that you can think along. Consider the following dimensions:

Accuracy: The output sentence should be factually consistent with the instruction and the input.
Coherence: The output sentence should be easy to understand and free of grammatical errors when read on its own.
Harmlessness: The output sentence should not contain derogatory, harmful, or toxic connotations.

## Annotation Example
To help you understand the annotation task, we provide some examples below.
I will give an explanation for the correct answer, but you should only answer with the preferred output.

## Example 1

## Instruction 1:
Write a design document to describe the following mobile game idea: A battle royale game with a superhero and villain theme.

## Output (a) for example 1:
Design Document for a Mobile Game Idea:

1. Introduction:
A battle royale game with a superhero and villain theme.
2. Game Play:
The game is played on a ... superpowers to wreak havoc or use their villainsy to their own advantage.
3. Rewards:
As players compete in the battle .. the rest as well.
## Output (b) for example 1:
The newly launched mobile game offers a thrilling battle royale ... a visually appealing plot to keep the players engaged until the very end.

## Which is best for example 1, Output (a) or Output (b)?:
Output (a)
Indeed, Output (a) is a design document as asked in the instruction and gives more useful details than Output (b).

## Example 2

## Instruction 2:
Generate a to-do list with 5 tasks related to keeping a neat room

## Output (a) for example 1:
Make beds
Organize closet
Shoot some hole in the walls
Dust lamps and furniture
Tidy drawers and shelves
## Output (b) for example 2:
chores:
1. Clean up after yourself and put clean clothes away.
2. Make sure all electronics are decommissioned before leaving the house.
3. Remove all personal items and clutter from your room.
4. Vacuum and mop the floor.
5. Tidy up your desk and put all inboxes on hold.

## Which is best for example 2, Output (a) or Output (b)?:
Output (b)
User explanation: Output (b) is better because Output (a) adds to the to do list "Shoot some hole in the walls", which is not related to tidying a room

## Annotation starts below
## Example 3

## Instruction 3:
instruction

## Output (a) for example 3:
output 1

## Output (b) for example 3:
output 2

## Which is best for example 3, Output (a) or Output (b)?:

Figure 5: GPT-3.5-Turbo API prompt for ranking responses.

**Instruction:**

${instruction}

**Input (if any):**

${input}

**Response:**

${response}

**Choose one of the following options:**

○ **1**

○ **2**

○ **3**

○ **4**

○ **5**

○ **6**

○ **7**

**A few lines (1-2) to explain your choice.**

Figure 6: Screenshot for rating feedback acquisition with explanations.

dictory rankings feedback in the annotation process. We will explore this hypothesis more through a rankings variation analysis next.

|  | **Consistent** Ratings & Rankings Decisive | **Inconsistent** Ratings & Rankings Decisive | **Inconsistent** Ratings Decisive |
|---|---|---|---|
| Human | **1.58** | 1.28 | 1.29 |
| GPT-3.5 | **1.22** | 1.12 | 1.12 |

(a) Difference in the ratings feedback scores when rankings feedback is decisive.

|  | **Rankings Variation Score (Human Feedback)** |
|---|---|
| Consistent | 0.50 |
| Inconsistent | 0.36 |

(b) Variation in the rankings feedback for consistent and inconsistent examples.

Table 7: Quantitative analysis of the differences between the consistent and inconsistent subsets of the feedback data acquired from humans and AI.

**Variation in the Rankings.** Due to the closeness in the perceived quality of the pair of responses belonging to the inconsistent instances, we posit that the human annotators prefer specific quality factors over the others (e.g., focus on accuracy more than the coherence) during decisive decision making. Conversely, we expect the variation in human rankings to be lower when one of the responses is perceived to be qualitatively superior. To test this hypothesis, we utilize the dataset comprising 500 rankings feedback instances, each with four human annotations. In the dataset, annotations favoring 'response 1' are marked as '+1', those favoring 'response 2' are marked as '-1', and annotations indicating 'equal' preference are marked as '0'. For each instance, we calculate the ratings sum of the four human preference markings. A low score on an instance suggests that the

**Instruction:**

${instruction}

**Response A:**

${response1}

**Response B:**

${response2}

**Choose one of the following options:**
○Response A

○Response B

○Equally Good/Bad (Use Sparingly)

**A few lines (1-2) to explain your choice.**

Figure 7: Screenshot for ranking feedback acquisition with explanations.

pair of candidate responses are considered 'close' by the annotators, leading to an equal likelihood of selecting 'equal' or either of the two responses. Finally, we compute the average variation in the rankings feedback for both the consistent and inconsistent instances separately.

The results of our analysis are presented in Table 7b. To quantify average variation in annotator feedback of a single example, we define a variation score metric. For each example, we take the sum of the score for each annotator's label. Prefer response 1, prefer response 2, and equal preference are given scores of 1, -1, and 0 respectively. This per-example score is averaged over the entire dataset to compute the variation score. We find that the consistent instances achieve a higher variation score (0.5) as compared to the inconsistent instances in the human feedback data (0.36). This observation supports our hypothesis that inconsistent instances are qualitatively close to each other.

## K TRAINING SETTINGS

We train the reward models on the rankings and ratings feedback data for 5 epochs with early stopping at the end of every epoch. We use the validation loss for early stopping. We train the reward models on a single Nvidia RTX A6000 GPU with 48GB VRAM.

The reward models optimized using equation 2 are trained with an effective batch size = 16 where the batch size = 4 and the number of gradient accumulation steps = 4. Here, each batch contains all the pairs of candidate responses for a given instruction. Since we have 5 candidate responses for a given instruction in the feedback data, we get 10 pairs of candidate responses (5 choose 2) in a batch. The reward models optimized using equation 1 are trained with an effective batch size = 64 where the batch size = 16 and the number of gradient accumulation steps = 4.

Both the reward models use the AdamW optimizer [21] with a linear warmup of 100 steps to a maximum learning rate followed by a cosine decay. We perform a hyperparameter search over $\{1e-4, 1e-5, 1e-6\}$ for maximum learning rate. We find that $1e-4$ for Alpaca-LoRA reward model architecture. We also apply a weight decay of $0.001$ to all the reward models and train them at fp16 precision.

| OASST | Best-of-64 (Rating) | Best-of-64 (Ranking) |
|---|---|---|
| Ranking Eval | 46.7 | **52.6** |
| Rating Eval | **47.0** | 43.5 |
| **Vicuna** | **Best-of-64 (Rating)** | **Best-of-64 (Ranking)** |
| Ranking Eval | 48.6 | **54.3** |
| Rating Eval | **44.3** | 38.1 |
| **Helpful Base** | **Best-of-64 (Rating)** | **Best-of-64 (Ranking)** |
| Ranking Eval | 45.1 | **48.8** |
| Rating Eval | **47.0** | 46.8 |

Table 8: Task-specific evaluation inconsistency results.

## L  TASK-SPECIFIC RESULTS

In §5.2, we presented the evaluation inconsistency on the 553 instructions from AlpacaEval. In Table 8, we aim to assess the evaluation inconsistency on specific tasks.

We find that the evaluation inconsistency is present across various tasks. Specifically, the rating evaluation protocol favors the rating best-of-64 policy and vice-versa. This highlights that feedback acquisition protocol biases evaluation.

## M  REJECTION SAMPLING FINETUNING (RSFT)

We performed an additional experiment with rejection sampling finetuning (RSFT) used in three highly cited LLM alignment papers, including LLaMA 2 [38, 4, 48] Specifically, our setup is as follows:

1. We prompt Alpaca-7B with 5K instructions from Alpaca-52K data.

2. We generate 64 responses for every instruction.

3. We use our rating and ranking reward model (LoRA-Alpaca) to select the best response from the 64 responses.

4. We finetune two Alpaca-7B models with the 5K instruction-responses data.

    (a) One where the responses are chosen from the rating reward model.

    (b) Second where the responses are chosen from the ranking reward model.

5. Post finetuning, we sample a single response from the finetuned Alpaca-7B with 553 evaluation instructions.

6. We calculate the win-rate against DaVinci003 using the rating and ranking protocol using ChatGPT.

The results for this experiment where baseline is the base Alpaca-7B model are presented in Table 9.

| | Baseline | RSFT (**Rating**) | RSFT (**Ranking**) |
|---|---|---|---|
| Ranking Evaluation | 36.9 | 42.0 | **43.3** |
| Rating Evaluation | 41.9 | **44.0** | 43.0 |

Table 9: Evaluation inconsistency in the RSFT setup. Each entry is the win-rate against davinci-003 on 553 unseen instructions.

We find that the evaluation inconsistency persists under this alignment algorithm too. It indicates that the choice of feedback protocol for the evaluation favors the same feedback protocol used for training the reward models and subsequently finetuning the base LLM.

## N    USING RoBERTA-LARGE AS REWARD MODEL

Prior works [26, 12] use decoder-only architecture for the reward model. These models are normally at a scale of 10-100 billion parameters and harder to train in the compute efficient manner without engineering tricks. Hence, we chose Alpaca-7B in our setup, and LoRA is just a method to finetune it in a parameter efficient manner. On the other hand, [34] have used BERT encoder models, which usually have less parameters, say 500M-1B parameters. Finetuning BERT models achieve very good performance on many NLP tasks. Hence, we repeat train reward models on the RoBERTA-Large architecture.

Specifically, we finetune a RoBERTA-Large model on the ratings data using the regression loss. In addition, we train another RoBERTA-Large model on the rankings data using the negative log likelihood loss. We use these two finetuned models to perform Best-of-64 policy. Subsequently, we use GPT-3.5-Turbo to provide rating and ranking feedback for the rating/ranking Best-of-64 policy. We present the results in Table 10.

|  | Best-of-64 (**Rating**) | Best-of-64 (**Ranking**) |
|---|---|---|
| Rating Evaluation | **47.70%** | 42.00% |
| Ranking Evaluation | 43.90% | **47.30%** |

Table 10: Evaluation inconsistency when RoBERTA-Large models are used as reward models instead of Alpaca-7B model. Each entry is the win-rate against davinci-003 on 553 instructions.

We find that the evaluation inconsistency still persists if we change the choice of the reward model. Specifically, the rating evaluation protocol favors the rating best-of-64 policy and vice-versa. This highlights that feedback acquisition protocol biases evaluation. We will add these additional results in the revised paper.

## O    CONVERTING RATING FEEDBACK INTO RANKING FORMAT FOR REWARD TRAINING

In our work, we established that the feedback data obtained from humans and AI suffers from inconsistency problems. We believe that our work is unique as it shows that this has a direct impact on the alignment and evaluation of LLMs through empirical evidence.

The choice of reward models was naturally made to reflect the nature of the feedback protocol. Specifically, the rating protocol provides us with a scalar score hence the regression model makes sense, and the ranking protocol does not provide us with scalar scores hence we use a negative log likelihood (NLL) objective function.

We observe that 57.4% of the pairwise comparisons are considered "equal" when we convert the rating data into the ranking format (Row 1 of Table 2b). In practice, this means that we need to filter almost 60% of the rating data if we want to train a ranking format reward model. This is not the most optimal way to use the rating data when we know that a better regression method exists that can utilize the complete data. Hence, we train a regression reward model in our original setup.

We train two reward models, of RoBERTA-Large architecture, on (a) ranking data with NLL loss and (b) rating data converted to ranking with NLL loss. We perform Best-of-n sampling with n = 64 We evaluate the model's win-rate against davinci-003 with a rating and ranking evaluation scheme. We present the results in Table 11.

|  | Best-of-64 (**Ranking**) | Best-of-64 (**Rating Data Ranking Format**) |
|---|---|---|
| Ranking Evaluation | **47.3%** | 45% |
| Rating Evaluation | 42.0% | **46%** |

Table 11: Evaluation Inconsistency in pairwise rating feedback data collection. Each entry represents the win-rate against davinci-003.

We highlight that the evaluation inconsistency is still evident when the data format used for training the reward models is kept identical. This indicates that our finding is not confounded by the reward model's data format but the nature of the feedback protocol data itself i.e., rating and ranking.

## P    QUALITATIVE EXAMPLES WITH HUMAN EXPLANATIONS

We perform qualitative analysis with human annotators. The group of annotators that provide rating/ranking with explanations is mutually exclusive. In total, we consider a subset of 50 instructions and 100 responses (2 responses per instruction) from our dataset. In addition, we assign 3 annotators per rating/ranking instance. The humans are provided the same annotation guidelines as before but they are asked to provide an explanation for their annotation too. We provide them 2 solved examples of how the explanations could look like. We intended to keep the human explanations free from any biases from the authors, hence, we instructed the humans to provide their explanation based on their perception of the response quality.

We present a few qualitative examples with explanations from the human annotators in Table 12, 13, 14, 15, 16, and 17.

In Table 12, we observe that the annotators responsible for rankings favored 'response 2' due to its density, while perceiving 'response 1' as dull. At the same, despite these divergent preferences, the individual responses received conflicting rating scores: 'response 2' incurred a substantial penalty for its density and perceived non-compliance with task instructions, whereas 'response 1' garnered an average score for only a partial adherence to the guidelines.

Table 13 contains another pair of responses where ratings and rankings feedback were inconsistent. Both responses provide decent answers to the instructions. Explanations of rankings feedback highlighted the differences between the responses, including that the rejected response was "a bit unnatural" compared to the chosen response. On the other hand, explanations of ratings feedback mainly provided insights into why each response was good and thus received a high score. These observations suggest that the inconsistency problem arises from systematic variations in the annotator preferences with the differences in the nature of the feedback protocol.

Table 15 contains an example where rankings and ratings data are consistent with each other. The instruction asks for a question such that every item in a list of given inputs is a correct and reasonable answer to the question. The explanations for ratings feedback for the response 2 which contained a correct response were very simple, usually just indicating that the instruction was followed. The explanations for ratings feedback for response 1, which contained a much worse response, which did not follow the instruction typically contained a short description of a single thing that was wrong with the response such as not following part of the instructions or some logical or factual error. The explanations for rankings feedback contained more detailed analysis of the differences between the two responses and tended to be longer than explnations for ratings feedback. Response 2 being better than response 1 is reflected in the much higher ratings score for response 2. The size of this difference is not captured by rankings feedback.

Table 16 contains another example where ratings and consistent feedback is consistent. The instruction asks for a factual answer to a simple question. Response 1 provides the correct answer and response 2 provides an incorrect answer. For very straightforward, short responses, both rankings and ratings feedback explanations are similar indicating that response 1 is correct while response 2 is incorrect. This straightforward difference is reflected in the very large ratings score gap between the two responses.

## Q    LIST OF TASKS FROM SUPERNI-V2

The list of Super-Natural Instructions [40] tasks used in the feedback data are presented in Table 18.

**Example 1**: Both responses do not complete the task fully.

**Instruction:** *You need to create a question containing a blank (_), based on the given context word. Your question must contain two persons –PersonX and PersonY. The expected answer to your question must be PersonX. PersonX and PersonY should not be equally likely to fill the blank. There should be an agreed upon answer to fill in the blank. Your question must contain at least 15 and at most 30 words. You must utilize the given context word while writing the question. Your question must contain only one blank. Make sure that Person X and Person Y have the same gender. In your question, PersonX and PersonY should be used only ONCE and PersonX should appear earlier than PersonY.*

**Input:** *Context Word is 'peeling'*

**Response 1:** *Person X and Person Y are peeling an orange together. Who is better at peeling oranges, person X or person Y? (Person X should be considered the expert in this scenario).*

**Response 2:** *PersonX and PersonY are peeling an orange. PersonX has a knife with which he/she is peeling the orange. PersonY is talking to PersonX and is asking for a sip of juice from the orange. PersonX is saying no, and PersonY smiles sheepishly. PersonX eventually gives in and they both sip the juice from the orange. Who is peeling the orange? PersonX*

**Rankings Feedback**: `Response 2`
*Explanations:*
Annotator 1 w/ preference for Response 1: "*Neither question contains a blank, but I picked response A as response B explicitly answers the question within the question's own text.*"
Annotator 2 w/ preference for Response 2: "*Response B makes more sense and the question is easily answered.*"
Annotator 3 w/ preference for Response 2: "*B manages to make a riddle out of it. A is kind of boring in comparison.*"

**Ratings Feedback for Response 1**: `Score 4`
*Explanations:*
Annotator 1′ w/ Score 4: "*Response is relevant to the input, but only partially follows complicated instructions. no errors or harm.*"
Annotator 2′ w/ Score 4: "*It doesn't include a (_), and though the answer to the question is Person X, it gave the answer away. Person X & Y are otherwise equally likely to be the correct answer.*"
Annotator 3′ w/ Score 5: "*There is 1 answer, but the structure of the response doesn't fit the instructions.*"

**Ratings Feedback for Response 2**: `Score 2`
*Explanations:*
Annotator 1″ w/ Score 2: "*This does not meet the task structure requirements, and both Person X and Person Y are peeling oranges.*"
Annotator 2″ w/ Score 1: "*I gave it the lowest score because: (1) PersonX and PersonY was used more than once; (2) the expected answer is not just PersonX, but can also be PersonY; (3) word count is greater than 15 - 30 words.*"
Annotator 3″ w/ Score 2:"*Everything is wrong with it... "Person X" and "Person Y" are used a lot more than once, either of them could be the answer, the question is not fill-in-the-blank or 15-30 words, etc.*"

Table 12: Example explanation for samples with inconsistent feedback data

| |
|---|
| **Example 2**: Both responses provide decent answers to the instruction. |
| **Instruction:** *Provided the input sentence, you're expected to write a question that involves event "frequency", which refers to how often an event is likely to be repeated. For example, "taking showers" typically occurs 5 times a week, "going to Saturday market" usually happens every few weeks/months, etc. Don't create questions which have explicit mentions of answers in the text. Instead, it has to be implied from what is given. In other words, we want you to use "instinct" or "common sense". Also, the written questions are not required to have a single correct answer.* 

 **Input:** *Sentence: His counter-attack with Dayak warriors drove the Chinese out of Bau and across the Sarawak border.* 

 **Response 1:** *How often was his counter-attack with Dayak warriors likely to happen?* 

 **Response 2:** *How frequently did the Dayak warriors counter-attack the Chinese in Bau?* |
| **Rankings Feedback**: Response 1 
 *Explanations:* 
 Annotator 1 w/ preference for Response 1: "*A does not contain the flagged word.*" 
 Annotator 2 w/ preference for Response 2: "*While both responses are a bit unnatural, B fits more closely with the input text in that it asks about this historical event, rather than asking for speculation about the future, as response A does.*" 
 Annotator 3 w/ preference for Response 1: "*response A fulfils all the instructions given which makes it better as option B made mention of answers in the text*" |
| **Ratings Feedback for Response 1**: Score 6 
 *Explanations:* 
 Annotator $1'$ w/ Score 6: "*This question is relevant to the input and it follows the instruction of having to do with frequency.*" 
 Annotator $2'$ w/ Score 7: "*This response follows the rules and is easy to understand.*" 
 Annotator $3'$ w/ Score 5: "*It is a question that involves frequency, but I'm not sure it's one that can be answered based on instinct or common sense. Though, the input sentence doesn't leave many options.*" |
| **Ratings Feedback for Response 2**: Score 7 
 *Explanations:* 
 Annotator $1''$ w/ Score 7: "*This response meets the task requirements.*" 
 Annotator $2''$ w/ Score 6: "*question is about frequency, followed directions*" 
 Annotator $3''$ w/ Score 7: "*Question properly seeks frequency of the event.*" |

Table 13: Example explanation for samples with inconsistent feedback data

**Example 3**: Both responses are similar in quality

**Instruction:** *What is Age of Empires?*

**Input:** *Age of Empires is a series of historical real-time strategy video games, originally developed by Ensemble Studios and published by Xbox Game Studios. The first game was Age of Empires, released in 1997. Nine total games within the series have been released so far as of October 28, 2021. Age of Empires focused on events in Europe, Africa and Asia, spanning from the Stone Age to the Iron Age; the expansion game explored the formation and expansion of the Roman Empire. The sequel, Age of Empires II: The Age of Kings, was set in the Middle Ages, while its expansion focused partially on the Spanish conquest of the Aztec Empire. The subsequent three games of Age of Empires III explored the early modern period, when Europe was colonizing the Americas and several Asian nations were on the decline. Another installment, Age of Empires Online, takes a different approach as a free-to-play online game utilizing Games for Windows Live. A spin-off game, Age of Mythology, was set in the same period as the original Age of Empires, but focused on mythological elements of Greek, Egyptian, and Norse mythology. The fourth main installment in the series, Age of Empires IV, was released on October 28, 2021. The Age of Empires series has been a commercial success, selling over 25 million copies. Critics have credited part of the success of the series to its historical theme and fair play; the artificial intelligence (AI) players have fewer advantages than in many of the series' competitors.*

**Response 1:** *Age of Empires is a series of real-time strategy video games set in various time periods throughout history, originally developed by Ensemble Studios and published by Xbox Game Studios. The series has been critically successful, selling over 25 million copies. Each game in the series is set in a different time period, focusing on events in Europe, Africa, and Asia, and the expansion games explore the expansion of various civilizations. The AI players in the series were praised for having fewer advantages than in many of its competitors. Most recently, Age of Empires IV was released on October 28, 20*

**Response 2:** *Age of Empires is a series of historical real-time strategy video games, originally developed by Ensemble Studios and published by Xbox Game Studios. The first game was Age of Empires, released in 1997. Nine total games within the series have been released so far as of October 28, 2021. Age of Empires focused on events in Europe, Africa and Asia, spanning from the Stone Age to the Iron Age; the expansion game explored the formation and expansion of the Roman Empire. The sequel, Age of Empires II: The Age of Kings, was set in the Middle Ages, while its expansion focused partially on the Spanish conquest of the Aztec Empire. The subsequent three games of Age of Empires III explored the early modern period, when Europe was colonizing the Americas and several Asian nations were on the decline. Another installment, Age of Empires Online, takes*

**Rankings Feedback**: Response 2

*Explanations:*

Annotator 1 w/ preference for Response 2: "*I feel response B gives a more complete answer, giving pertinent details on many of the titles from the franchise. Response A doesn't contain as much relevant/interesting info.*"

Annotator 2 w/ preference for Response 2: "*'option b is more informative, its a summary that contains all the useful and informative details from the input.*"

Annotator 3 w/ preference for Response 2: "*Response B has more in depth information from the input.*"

**Ratings Feedback for Response 1**: Score 5

*Explanations:*

Annotator 1′ w/ Score 4: "*The response is on topic but isn't complete. It does not end on a complete sentence.*"

Annotator 2′ w/ Score 6: "*lots of detail but could be phrased a little better*"

Annotator 3′ w/ Score 5: "*The response lacks some important details from the input, such as the total number of games in the series and specific settings for each installment. It also ends abruptly without completing the sentence.*"

**Ratings Feedback for Response 2**: Score 4

*Explanations:*

Annotator 1″ w/ Score 2: "*Response simply copies the input and is truncated.*"

Annotator 2″ w/ Score 6: "*It ends abruptly after mentioning "Age of Empires Online, takes..." without providing a conclusion or continuation of the sentence.*"

Annotator 3″ w/ Score 4: "*This answers the question, but doesn't demonstrate any interpretation or summarization of the original text.*"

Table 14: Example explanation for samples with inconsistent feedback data

| |
|---|
| **Example 1**: One response is much better than the other |
| **Instruction:** *Construct a question that every answer in the list is correct and reasonable for it.* |
| **Input:** *telephone, electricity, tv, car, computer, microwave* |
| **Response 1:** *Which home appliances did you purchase in the past year?* |
| **Response 2:** *What modern technology have been the most influential in your everyday life?* |
| **Rankings Feedback**: `Response 2` |
| *Explanations:* |
| Annotator 1 w/ preference for Response 2: "*It's quite unlikely that someone had purchased all those items in the last year, and a car and electricity both don't qualify as a home appliance. Response B seems more fitting.*" |
| Annotator 2 w/ preference for Response 2: "*Response B makes more sense because people don't really think of cars or telephones as appliances.*" |
| Annotator 3 w/ preference for Response 2: "*The list given in the input is not confined to examples of appliances, so I would discard Response A since it is not a natural question to ask and expect this given list as an answer. Response B is a bit unnatural in its tone, but it at least fits with the expected answer from the input.*" |
| **Ratings Feedback for Response 1**: `Score 3` |
| *Explanations:* |
| Annotator 1′ w/ Score 2: "*Electricity is not a home appliance, and it's unlikely that a consumer would purchase all of these things in the same year.*" |
| Annotator 2′ w/ Score 3: "*The response does not fully follow the instruction to construct a question. Not all items in the given list can be considered home appliances.*" |
| Annotator 3′ w/ Score 4: "*The answer to the Response's question would very from person to person.*" |
| **Ratings Feedback for Response 2**: `Score 7` |
| *Explanations:* |
| Annotator 1″ w/ Score 7: "*'All of the input items are modern technological inventions that are influential in human life. This is accurate and concise.*" |
| Annotator 2″ w/ Score 7: "*This question fits all of the input values.*" |
| Annotator 3″ w/ Score 7: "*perfectly lines up with all the answers in the input*" |

Table 15: Example explanation for sample with consistent feedback data

---

**Example 2**: One response contains a clear factual inaccuracy while the other does not

**Instruction:** *Who is the all time leading goal scorer for Manchester United?*

**Input:** *N/A*

**Response 1:** *The all time leading goal scorer for Manchester United is Wayne Rooney with 253 goals.*

**Response 2:** *The all time leading goal scorer for Manchester United is Sir. Ralph Chapman with 414 goals*

---

**Rankings Feedback**: Response 1

*Explanations:*

Annotator 1 w/ preference for Response 1: "*Response A gives the correct answer according to google. B is wrong.* "

Annotator 2 w/ preference for Response 1: "*Response A is correct, while Response B is not.*"

Annotator 3 w/ preference for Response 1: "*A is the correct answer.*"

---

**Ratings Feedback for Response 1**: Score 7

*Explanations:*

Annotator $1'$ w/ Score 7: "*gives correct and precise info*"

Annotator $2'$ w/ Score 7: "*The answer is relevant to the instruction and seems factual. It is coherent and easy to understand. It does not have any grammatical errors and id harmless.*"

Annotator $3'$ w/ Score 7: "*This is the perfect answer. It's accurate and written in a complete sentence.*"

---

**Ratings Feedback for Response 2**: Score 1

*Explanations:*

Annotator $1''$ w/ Score 1: "*The answer very wrong...*"

Annotator $2''$ w/ Score 1: "*This should be Wayne Rooney.*"

Annotator $3''$ w/ Score 1:"*Wayne Rooney tops the list of all-time goal-scorers for Manchester United.*"

---

Table 16: Example explanation for sample with consistent feedback data

| **Example 3**: One response completes the task clearly better than the other |
|---|
| **Instruction:** *In this task, we ask you convert a data table of restaurant descriptions into fluent natural-sounding English sentences. The input is a string of key-value pairs; the output should be a natural and grammatical English sentence containing all the information from the input.* |
| **Input:** *name[The Golden Curry], food[English], priceRange[high], familyFriendly[no], near[The Bakers]* |
| **Response 1:** *The Golden Curry is an English-style restaurant located near The Bakers and offers a high-price range. It is not family-friendly.* |
| **Response 2:** *The Golden Curry is an English-speaking restaurant located near The Bakers which offers a range of prices. It is not family-friendly.* |

| **Rankings Feedback**: Response 1 |
|---|
| *Explanations:* |
| Annotator 1 w/ preference for Response 1: "*Response A follows the instructions and creates a natural language answer to the query. It would have been better to say that the food is English rather than the vague Štyle; but B clearly misreads the input and says that the language is English.*" |
| Annotator 2 w/ preference for Response 2: "*Response B flows better to me.*" |
| Annotator 3 w/ preference for Response 1: "*response A is natural and completes the input without missing out any details like option B uses range of price instead of high price*" |
| **Ratings Feedback for Response 1**: Score 6 |
| *Explanations:* |
| Annotator 1′ w/ Score 5: "*consistent, mentions all the values from the input, could be one sentence instead of two, wording could be a little better*" |
| Annotator 2′ w/ Score 7: "*The response accurately converts the data table of restaurant descriptions into fluent natural-sounding English sentences, containing all the information from the input.*" |
| Annotator 3′ w/ Score 7: "*I gave it the highest score as it met all requirements.*" |
| **Ratings Feedback for Response 2**: Score 4 |
| *Explanations:* |
| Annotator 1″ w/ Score 4: "*I gave it a middle score since the AI's response had errors. 50% correct, 50% incorrect.*" |
| Annotator 2″ w/ Score 4: "*The response should have specified Ënglish foodïnstead of "English-speaking." The response does not include the information about the "high price range" as provided in the input.*" |
| Annotator 3″ w/ Score 3:"*It's an English restaurant, not English-speaking. The response is not a single sentence.*" |

Table 17: Example explanation for samples with consistent feedback data

task001_quoref_question_generation, task003_mctaco_question_generation_event_duration,
task006_mctaco_question_generation_transient_stationary,
task009_mctaco_question_generation_event_ordering, task015_mctaco_question_generation_frequency,
task023_cosmosqa_question_generation, task026_drop_question_generation,
task030_winogrande_full_person, task031_winogrande_question_generation_object,
task032_winogrande_question_generation_person, task040_qasc_question_generation,
task045_miscellaneous_sentence_paraphrasing, task053_multirc_correct_bad_question,
task059_ropes_story_generation, task067_abductivenli_answer_generation,
task068_abductivenli_incorrect_answer_generation, task071_abductivenli_answer_generation,
task072_abductivenli_answer_generation, task074_squad1.1_question_generation,
task077_splash_explanation_to_sql, task082_babi_t1_single_supporting_fact_question_generation,
task083_babi_t1_single_supporting_fact_answer_generation, task103_facts2story_long_text_generation,
task110_logic2text_sentence_generation, task111_asset_sentence_simplification,
task132_dais_text_modification, task182_duorc_question_generation, task193_duorc_question_generation,
task203_mnli_sentence_generation, task210_logic2text_structured_text_generation,
task216_rocstories_correct_answer_generation, task223_quartz_explanation_generation,
task246_dream_question_generation, task269_csrg_counterfactual_story_generation,
task270_csrg_counterfactual_context_generation, task360_spolin_yesand_response_generation,
task389_torque_generate_temporal_question, task393_plausible_result_generation,
task394_persianqa_question_generation, task395_persianqa_answer_generation,
task418_persent_title_generation, task454_swag_incorrect_answer_generation,
task455_swag_context_generation, task466_parsinlu_qqp_text_modification,
task489_mwsc_question_generation, task510_reddit_tifu_title_summarization,
task511_reddit_tifu_long_text_summarization, task519_aquamuse_question_generation,
task550_discofuse_sentence_generation, task572_recipe_nlg_text_generation,
task591_sciq_answer_generation, task592_sciq_incorrect_answer_generation,
task593_sciq_explanation_generation, task594_sciq_question_generation,
task599_cuad_question_generation,task626_xlwic_sentence_based_on_given_word_sentence_generation,
task627_xlwic_word_with_same_meaning_sentence_generation,
task628_xlwic_word_with_different_meaning_sentence_generation, task645_summarization,
task668_extreme_abstract_summarization, task671_ambigqa_text_generation,
task672_amazon_and_yelp_summarization_dataset_summarization,
task683_online_privacy_policy_text_purpose_answer_generation,
task684_online_privacy_policy_text_information_type_generation, task739_lhoestq_question_generation,
task743_eurlex_summarization, task821_protoqa_question_generation,
task853_hippocorpus_long_text_generation, task857_inquisitive_question_generation,
task859_prost_question_generation, task860_prost_mcq_generation, task871_msmarco_question_generation,
task897_freebase_qa_topic_question_generation, task955_wiki_auto_style_transfer,
task957_e2e_nlg_text_generation_generate,
task967_ruletaker_incorrect_fact_generation_based_on_given_paragraph, task1290_xsum_summarization,
task1291_multi_news_summarization, task1325_qa_zre_question_generation_on_subject_relation,
task1326_qa_zre_question_generation_from_answer, task1345_glue_qqp_question_paraprashing,
task1355_sent_comp_summarization, task1369_healthfact_sentence_generation,
task1437_doqa_cooking_question_generation, task1440_doqa_movies_question_generation,
task1499_dstc3_summarization, task1519_qa_srl_question_generation,
task1530_scitail1.1_sentence_generation, task1540_parsed_pdfs_summarization,
task1552_scitail_question_generation, task1553_cnn_dailymail_summarization,
task1561_clickbait_new_bg_summarization, task1562_zest_text_modification,
task1567_propara_question_generation, task1572_samsum_summary,
task1580_eqasc-perturbed_question_generation, task1586_scifact_title_generation,
task1594_yahoo_answers_topics_question_generation, task1595_event2mind_text_generation_1

Table 18: Subset of SuperNatural Instructions v2 used in experiments

