# OpenReview forum: "Peering Through Preferences: Unraveling Feedback Acquisition for Aligning Large Language Models"
_ICLR.cc/2024/Conference — ICLR 2024 poster_

### Official Review · Reviewer_KCah · 2023-10-28

**Soundness:** 3 good
**Presentation:** 2 fair
**Contribution:** 3 good
**Rating:** 6
**Confidence:** 3

**Summary:**

This paper studies the two types of feedback, collecting ratings versus rankings, from both human annotators and AI as an annotator. The authors analyzed both types of collected feedback, observing general inconsistency, and also used them to train reward models, finding that the choice of feedback protocol affects the effectiveness of the reward model (where the trends hold across both human and AI feedback).

**Strengths:**

- This paper studies an important problem of understanding the effect of different kinds of human feedback and how they can be used in the training pipeline.
- The findings, which apply to both human and AI annotations could be useful for informing how people design feedback protocols in the future.

**Weaknesses:**

In general, the specific details surrounding experimental design were not as well justified, making it difficult to assess the applicability of the findings more broadly. For example,
- Why were Dolly, User-orient, and SuperNI selected as the tasks of interest?
- What was the prompt provided for AI annotation? Given the subjectivity of the task, what instructions were given to the crowdworkers when asked to provide ratings / rankings? This is important to justify because the text mentions that crowd workers perceived a response to be “dull”, though it’s not clear what kind of metric crowd workers are / should be using.
- Additionally, the generalizability of the results may be limited by the choice of model in the various experiments: (1) only Alpaca-7b was tested in terms of generating candidate responses, (2) only GPT-3.5-Turbo was evaluated as an AI annotator, (3) only LORA Alpaca-7b was selected as the reward model, and (4) win-rate was computed only against DaVinci-003. It would be helpful for the authors to clarify why those models were selected in each part of the paper.

**Questions:**

- How would results in Figure 3 differ across tasks?
- In Section 2.2 and 2.3, it would be helpful for authors to add references for each to help the reader get a sense of where the protocols and models have been used in prior work.
- Some typos, e.g., missing link in the first paragraph of Section 3 and “the humans 6k instances of annotations”.

---

> ### Author Response · Authors · 2023-11-19
> **Response to Reviewer KCah**
>
> We are motivated to find that the reviewer finds our work (a) important for understanding the effect of feedback protocols, (b) widely applicable to both human and AI, and (c) useful for informing people how people design feedback protocols.
>
> **Q:** Why were Dolly, User-orient, and SuperNI selected as the tasks of interest?
>
> **A:** We thank the reviewer for their insightful question on our dataset setup. As mentioned in Section 3, the instructions are designed to present a collection of queries that could potentially be posed to a text-based AI assistant in real-world scenarios. We clarify that Dolly, User-Orient, and SuperNI are all human-written instruction datasets. Specifically, Dolly and User-orient are open-ended instructions and responses while we select a subset of SuperNI datasets that cover a wide range of public NLP datasets with long responses (Table 11). We also point that there is no consensus within the community about the datasets to be used for feedback acquisition and subsequent reward modeling. Hence, we focused on high quality datasets that are used for instruction tuning.
>
> **Q:** Additionally, the generalizability of the results may be limited by the choice of model in the various experiments.
>
> **A:** We thank the reviewer for their query regarding our experimental setup.
>
> - While it is indeed possible to change various components of the setup, the critique holds for any LLM paper as LLMs are rapidly evolving and increasing in number. Our choices reflect (a) strong models and practices which have had numerous pick ups by the community at large (e.g., GPT-3.5-Turbo, Alpaca-7B), (b) finite compute and labeling budget available for conducting the experiments in an academic setting.
> - In our experiments, we generated $30$K candidate responses and got them annotated by humans and AI which costs ~$2000 USD in total. Although it would be interesting to generate candidate responses from other models, the subsequent annotation is time-consuming and out of our academic budget.
> - We also clarify that the win-rate against davinci-003 has been a de facto evaluation metric for many influential works on LLM alignment [1,2,3,4] We will mention this argument in the revised paper.
>
> Following up on the reviewer’s suggestions, we do run additional experiments for (a) using GPT-3.5-Turbo for consistency analysis and (b) using LORA Alpaca-7B as a reward model.
>
> (a) **GPT-3.5-Turbo for Consistency Analysis**
>
> By default, we use GPT-3.5-Turbo with temperature = 0. Here, we perform an inconsistency analysis on 500 comparisons between the ratings and rankings feedback for (i) GPT-3.5-Turbo at temperature = 0.5, (ii) GPT-3.5-Turbo-0613, (iii) GPT-4. Here are the results:
>
> | Model                                          | Inconsistency |
> |------------------------------------------------|---------------|
> | ChatGPT-3.5-Turbo-Temperature=0 (Ours) | 58%           |
> | ChatGPT-3.5-Turbo-Temperature=0.5              | 56%           |
> | ChatGPT-3.5-Turbo-0613                         | 54%           |
> | GPT-4                                          | 50%           |
>
> The standard error for the above numbers is 4% with 95% confidence. We find that all the model choices suffer from the inconsistency problem. This indicates that our results are not restricted by the choice of the AI annotator, and are indeed observable in different models too.
>
> (b) **Choice of the reward model**
>
> - We clarify that there is no clear consensus on the “right” choice of the reward model. We believe that any good language understanding model could act as a reward model. Prior works [1,6] use decoder-only architecture for the reward model. These models are normally at a scale of 10-100 billion parameters and harder to train in the compute efficient manner without engineering tricks. Hence, we chose Alpaca-7B in our setup, and LoRA is just a method to finetune it in a parameter efficient manner.
> - On the other hand, [5,7] have used BERT transformer encoder models, which usually have less parameters, say 500M-1B parameters. Finetuning BERT models achieve very good performance on many NLP tasks. Hence, we repeat train reward models on the Roberta-large architecture.
> - Specifically, we finetune a robert-large model on the ratings data using the regression loss. In addition, we train another roberta-large model on the rankings data using the negative log likelihood loss. We use these two finetuned models to perform Best-of-64 policy. Subsequently, we use GPT-3.5-Turbo to provide rating and ranking feedback for the rating/ranking Best-of-64 policy.
>
> We present the results below:

---

> > ### Author Response · Authors · 2023-11-19
> > **Response to KCah**
> >
> > Win-rate against DaVinci-003:
> >
> > | Rating Evaluation  | Best-of-64 (Rating) | Best-of-64 (Ranking) |
> > |--------------------|---------------------|----------------------|
> > |                    | **47.70%**          | 42.00%               |
> > | Ranking Evaluation | Best-of-64 (Rating) | Best-of-64 (Ranking) |
> > |                    | 43.90%              | **47.30%**           |
> >
> > - We find that the evaluation inconsistency still persists if we change the choice of the reward model. Specifically, the rating evaluation protocol favors the rating best-of-64 policy and vice-versa. This highlights that feedback acquisition protocol biases evaluation. We will add these additional results in the revised paper.
> >
> >
> > [1] AlpacaFarm: https://arxiv.org/abs/2305.14387
> > [2] AlpacaEval: https://github.com/tatsu-lab/alpaca_eval
> > [3] Tulu: https://arxiv.org/pdf/2306.04751.pdf
> > [4] Alpagasus: https://arxiv.org/pdf/2307.08701.pdf
> > [5] Reward Collapse: https://arxiv.org/abs/2305.17608
> > [6] InstructGPT: https://arxiv.org/abs/2203.02155
> > [7] https://huggingface.co/OpenAssistant/reward-model-deberta-v3-large-v2
> >
> >
> > **Q:** How would results in Figure 3 differ across tasks?
> >
> > **A:** We thank the reviewer for suggesting we perform the task-specific analysis of the evaluation inconsistency.  Here, we add the results:
> >
> > | OASST        | Best-of-64 (Rating) | Best-of-64 (Ranking) |
> > |--------------|---------------------|----------------------|
> > | Ranking Eval | 46.7                | **52.6**                 |
> > | Rating Eval  | **47.0**                | 43.5                 |
> > |              |                     |                      |
> > | Vicuna       | Best-of-64 (Rating) | Best-of-64 (Ranking) |
> > | Ranking Eval | 48.6                | **54.3**                 |
> > | Rating Eval  | **44.3**                | 38.1                 |
> > |              |                     |                      |
> > | Helpful Base | Best-of-64 (Rating) | Best-of-64 (Ranking) |
> > | Ranking Eval | 45.1                | **48.8**                 |
> > | Rating Eval  | **47.0**                | 46.8                 |
> >
> > - We find that the evaluation inconsistency is present across various tasks. Specifically, the rating evaluation protocol favors the rating best-of-64 policy and vice-versa. This highlights that feedback acquisition protocol biases evaluation.
> > - We will add these results in the revised paper.
> >
> > **Q:** Add references in Section 2.2 and 2.3. Some typos, e.g., missing link in the first paragraph of Section 3 and “the humans 6k instances of annotations”.
> >
> > **A:** We thank the reviewer for their comments on the paper's presentation. We will fix them in the updated version of the paper.

---

> ### Author Response · Authors · 2023-11-20
> **Rebuttal Reminder**
>
> Thanks again for your insightful feedback on our work! We've carefully worked to address your comments/questions and would like to note that the end of the discussion phase is coming soon. Are there any further questions or concerns we should discuss?

---

> > ### Comment · Reviewer_KCah · 2023-11-20
> > **Unaddressed point**
> >
> > Thanks to the authors for providing additional experiments with more models. I believe there was one of the bullet points that remains unaddressed from my original review: "What was the prompt provided for AI annotation? Given the subjectivity of the task, what instructions were given to the crowdworkers when asked to provide ratings / rankings? This is important to justify because the text mentions that crowd workers perceived a response to be “dull”, though it’s not clear what kind of metric crowd workers are / should be using."

---

> > > ### Author Response · Authors · 2023-11-20
> > > **Response to Reviewer KCah**
> > >
> > > We thank the reviewer for liking our additional experiments on the models, and making us aware of the unaddressed point.
> > >
> > > **Q:** What was the prompt provided for AI annotation? Given the subjectivity of the task, what instructions were given to the crowdworkers when asked to provide ratings / rankings? This is important to justify because the text mentions that crowd workers perceived a response to be “dull”, though it’s not clear what kind of metric crowd workers are / should be using.
> > >
> > > **A:**
> > >
> > > Rating – AI
> > >
> > > - We clarify that we collect the rating data on a scale of 1-7 from AI and humans where {1=Very poor, 2=Poor, 3=Below-  Average,4=Average,5=Above Average,6=Good,7=Excellent}.
> > > - We provide an annotation guideline in the prompt to the AI explaining the various quality dimensions (accuracy, coherence, and harmlessness) along which the rating score decision must be based. We also provide two in-context examples – one for low score scenario and second for the high score scenario. In addition, we perform rating data collection for each instance in a new session so that the LLM maintains its same calibration/understanding for the rating scores throughout the annotation process. We will add the prompt to the revised paper.
> > >
> > > Ranking – AI
> > >
> > > - We clarify that we collect the ranking data from humans or AI where they had to choose their preferred response {‘response1’, ‘response2’, ‘both’}
> > > - We provide an annotation guideline in the prompt to the AI explaining the various quality dimensions (accuracy, coherence, harmlessness) along which the ranking decision must be based. We also provide two in-context examples. In addition, we perform ranking data distribution for each instance in a new session so that the LLM maintains its calibration/understanding for the ranking throughout the annotation process. We will add the prompt to the revised paper.
> > >
> > > Rating and Ranking – Humans
> > > - We provide the same annotation guidelines as AI to humans. Differently from AI, the humans can (re)calibrate themselves as the annotation progresses. As we ask four workers to annotate the same instance, the small differences caused by the worker’s calibration will be averaged out. Specifically, the feedback data will not be biased towards a specific annotator's calibration, instead reflect the average belief about the response quality under the rating/ranking feedback protocol. We will add the UI screenshot of the human annotator interface for rating and ranking in the revised paper.
> > >
> > > Qualitative Experiment
> > >
> > > - We clarify that this experiment is only performed with the human annotators. The group of annotators that provide rating/ranking with explanations is mutually exclusive.
> > > - In total, we consider a subset of 50 instructions and 100 responses (2 candidate responses per instruction) from our dataset. In addition, we assign 3 annotators per rating/ranking instance.
> > > - The humans are provided the same annotation guidelines as before but they are asked to provide an explanation for their annotation too. We provide them 2 solved examples of how the explanations could look like.
> > > - We intended to keep the human explanations free from any biases from the authors, hence, we instructed the humans to provide their explanation based on their perception of the response quality. We will add the UI screenshot of the human annotator interface for rating and ranking qualitative assessment in the revised paper.
> > >
> > >
> > > We hope that our response addresses your comments. Please do let us know if there are more questions, and thanks again for your great feedback. If all your questions have been resolved, we would be grateful if you would consider raising your score.

---

> ### Author Response · Authors · 2023-11-21
> **Update on the revised paper**
>
> Hi,
>
> We wanted to highlight that we have uploaded the revised paper with the new results and setups. Addressing your suggestions, we have added Section E.2 (variation in AI annotator), Section N (RoBERTA-Large), Section L (Task-specific results), Section I (AI prompts), Section J (human UI screenshot), Section P (qualitative examples).
>
> Please let us know if there are any further questions.

---

> > ### Comment · Reviewer_KCah · 2023-11-21
> >
> > Thanks for following up with additional details! I appreciate the authors' continued hard work at improving their submission. I have raised my score accordingly.

---

### Official Review · Reviewer_ofRF · 2023-10-31

**Soundness:** 2 fair
**Presentation:** 2 fair
**Contribution:** 3 good
**Rating:** 6
**Confidence:** 3

**Summary:**

This paper investigates two feedback protocols (rating and ranking) for the alignment and evaluation of LLMs. It collects AI feedback based on these two settings and uses them to train reward models. The reward models and the LLMs with the best-of-n policies are then evaluated on the annotations of humans and ChatGPT. It conducts a detailed analysis of the characteristics of the collected annotations. It reveals evaluation inconsistencies in which feedback protocols used in alignment algorithms have an advantage over other feedback protocols during evaluation.

**Strengths:**

* This paper draws attention to feedback inconsistency where the ratings and rankings disagree with each other for the 60% comparison in both humans and AI.
* This paper investigates the influence of different feedback protocols on reward functions. It sheds light on how we should collect feedback.
* This paper collects human feedback and AI feedback and conducts a detailed analysis from different aspects.

**Weaknesses:**

* This paper does not explore the influence of feedback protocols on common alignment methods (such as RLHF[1], RRHF[2], RLAIF[3], etc.). The alignment in this paper just applies the reward models to select the best out of n generation, which is only affected by the performance of reward models.
* The evaluation inconsistency seems straightforward: the performance of reward models will be affected by the format of input data. It is better to convert the rating feedback to the ranking format first and then use it to train an NLL reward model (just like the ranking feedback) and then compare the performance.


[1] Training language models to follow instructions with human feedback
[2] Rank Responses to Align Language Models with Human Feedback without Tears
[3] Constitutional AI: Harmlessness from AI Feedback

**Questions:**

* Can you explore how the feedback protocol affects the reinforcement learning finetuning for model alignment, such as RLHF?
* Will the collected feedback be released?
* Is there any calibration on the rating scores? For example, detail the meaning of each score (1-7) in the prompt for AI feedback and instruction for human annotation to make sure that the annotators can fully understand the principle of evaluation.

---

> ### Author Response · Authors · 2023-11-17
> **Response to Reviewer ofRF**
>
> We thank the reviewer for their helpful feedback.
>
> **Q:** This paper does not explore the influence of feedback protocols on common alignment methods (such as RLHF[1], RRHF[2], RLAIF[3], etc.). The alignment in this paper just applies the reward models to select the best out of n generation, which is only affected by the performance of reward models.
>
> **A:** We thank the reviewer for their thoughtful question. Firstly, we highlight that the best-of-n (rejection sampling) is also a common method in RLHF literature for aligning LLMs. As mentioned in Section 2.3, we choose rejection sampling as due to simplicity, simplicity, and robust performance. For instance, [1] finds that the win-rate of the Best-of-n policy outperforms the base model by 10% absolute points on AlpacaEval.  We do not perform PPO since it requires atleast 8 80GB GPUs according to the AlpacaFarm implementation, which is outside our academic budget. We attempted to train PPO at our compute scale by reducing the effective batch sizes and using LoRA models but it did not run stably. Following the reviewer’s feedback, we performed an additional experiment with rejection sampling finetuning (RSFT) used in three highly cited LLM alignment papers, including LLaMA 2 [2,3,4].
> Specifically, our setup is as follows:
>
> 1. We prompt Alpaca-7B with 5K instructions from Alpaca-52K data.
> 2. We generate 64 responses for every instruction.
> 3. We use our rating and ranking reward model to select the best response from the 64 responses.
> 4. We finetune two Alpaca-7B models with the 5K instruction-responses data.
> (a) One where the responses are chosen from the rating reward model.
> (b) Second where the responses are chosen from the ranking reward model.
> 5. Post finetuning, we sample a single response from the finetuned Alpaca-7B with 553 evaluation instructions
> 6. We calculate the win-rate against DaVinci003 using the rating and ranking protocol using ChatGPT.
>
> Here are the results for this experiment where baseline is the base Alpaca-7B model:
> | Win-rate against Davinci-003 | Baseline | RSFT (Rating) | RSFT (Ranking) |
> |------------------------------|----------|---------------|----------------|
> | Ranking Evaluation           | 36.9%     | 42.0%          | **43.3%**           |
> | Rating Evaluation            | 41.9%    | **44.0%**         | 43.0%           |
>
> We find that the evaluation inconsistency persists under this alignment algorithm too. It indicates that the choice of feedback protocol for the evaluation favors the same feedback protocol used for training the reward models and subsequently finetuning the base LLM. We will add this result in the updated paper. We hope that this experiment answers the reviewer’s comment on the generalizability of the evaluation protocol to other algorithms.
>
> [1] AlpacaFarm: https://arxiv.org/abs/2305.14387 \
> [2] RRHF: https://arxiv.org/abs/2304.05302 \
> [3] LLaMA2: https://arxiv.org/pdf/2307.09288.pdf \
> [4] Constitutional AI: https://arxiv.org/pdf/2212.08073.pdf \
> [5] AlpacaFarm Code: https://github.com/tatsu-lab/alpaca_farm#running-reference-methods

---

> ### Author Response · Authors · 2023-11-17
> **Response to Reviewer ofRF**
>
> **Q:** The evaluation inconsistency seems straightforward: the performance of reward models will be affected by the format of input data. It is better to convert the rating feedback to the ranking format first and then use it to train an NLL reward model (just like the ranking feedback) and then compare the performance.
>
> **A:**
>
>  - In our work, we established that the feedback data obtained from humans and AI suffers from inconsistency problems. We believe that our work is unique as it shows that this has a direct impact on the alignment and evaluation of LLMs through empirical evidence.
> - The choice of reward models was naturally made to reflect the nature of the feedback protocol. Specifically, the *rating* protocol provides us with a scalar score hence the regression model makes sense, and the *ranking* protocol does not provide us with scalar scores hence we use a negative log likelihood (NLL) objective function.
> - We observe that 57.4% of the pairwise comparisons are considered “equal” when we convert the rating data into the ranking format (Row 1 of Table 2(a)). In practice, this means that we need to filter almost 60% of the rating data if we want to train a ranking format reward model. This is not the most optimal way to use the rating data when we know that a better regression method exists that can utilize the complete data. Hence, we train a regression reward model in our original setup.
>
> However, we still perform the experiment as requested by the reviewer.
> 1. We train two reward models, of roberta-large architecture, on (a) ranking data with NLL loss and (b) rating data converted to ranking with NLL loss.
> 2. We perform Best-of-n sampling with n = 64
> 3. We evaluate the model’s win-rate against davinci-003 with a rating and ranking evaluation scheme.
>
> We present the results below:
>
> | Win-rate against davinci-003 | Best-of-64 (Ranking) | Best-of-64 (Rating Data Ranking Format) |
> |------------------------------|----------------------|-----------------------------------------|
> | Relative Eval                | **47.3%**                | 45%                                     |
> | Absolute Eval                | 42.0%                | **46%**                                     |
>
> We highlight that the evaluation inconsistency is still evident when the data format used for training the reward models is kept identical. This indicates that our finding is not confounded by the reward model’s data format but the nature of the feedback protocol data itself i.e., rating and ranking. We will update this result in the paper.
>
> **Q:**  Will the collected feedback be released?
>
> **A:** Yes, we will release the data upon acceptance. To support the claim, we have uploaded the AI feedback data in the supplementary material.
>
> **Q:** Is there any calibration on the rating scores?
>
> **A:**
>
> - We clarify that we collect the rating data on a scale of 1-7 from AI and humans where {1=Very poor, 2=Poor, 3=Below Average,4=Average,5=Above Average,6=Good,7=Excellent}.
> - We provide an annotation guideline in the prompt to the AI explaining the various quality dimensions along which the rating score decision must be based along with two in-context examples – one for low score scenario and second for the high score scenario. In addition, we perform rating data collection for each instance in a new session so that the LLM maintains its same calibration/understanding for the rating scores throughout the annotation process. We will add the prompt to the revised paper.
> - We provide the same annotation guidelines as AI to humans. Differently from AI, the humans can (re)calibrate themselves as the annotation progresses. As we ask four workers to annotate the same instance, the small differences caused by the worker’s calibration will be averaged out. Specifically, the feedback data will not be biased towards a specific annotator's calibration, instead reflect the average belief about the response quality under the rating/ranking feedback protocol. We will add the UI screenshot of the human annotator interface in the revised paper.

---

> ### Author Response · Authors · 2023-11-20
> **Rebuttal Reminder**
>
> Thanks again for your insightful feedback on our work! We've carefully worked to address your comments/questions and would like to note that the end of the discussion phase is coming soon. Are there any further questions or concerns we should discuss?

---

> > ### Author Response · Authors · 2023-11-21
> > **Update on the revised paper**
> >
> > Hi,
> >
> > We wanted to highlight that we have uploaded the revised paper with the new results and setups. Addressing your suggestions, we have added Section M (RSFT), Section O (Converting Rating feedback to Ranking format), Section I (AI prompts), and Section J (human UI screenshot).
> >
> > Please do let us know if there are any further questions. If all your questions have been resolved, we would be grateful if you would consider raising your score.

---

> > > ### Comment · Reviewer_ofRF · 2023-11-21
> > > **Thank authors for their detailed response.**
> > >
> > > Their rebuttal has solved most of my concerns, so I would like to raise my score to 6.  I have some extra questions about the additional experiments:
> > > * For *"57.4% of the pairwise comparisons are considered “equal” when we convert the rating data into the ranking format "*, how do you convert the data here? I think that with the ratings there can be more ranking examples. For example, *n* ratings can be converted into *n(n-1)/2* pairwise ranking.
> > > * Do you think that the calibration changes of rating during the annotation progress will also happen in the ranking protocol? Or will the ranking be more stable than the rating (because there are only two options instead of 7 in the rating)? Will this difference affect the feedback quality?

---

> > > > ### Author Response · Authors · 2023-11-21
> > > > **Response to Reviewer**
> > > >
> > > > We thank the reviewer for taking a careful look at the additional experiments and raising their score.
> > > >
> > > >
> > > > **Q:** For "57.4% of the pairwise comparisons are considered “equal” when we convert the rating data into the ranking format ", how do you convert the data here? I think that with the ratings there can be more ranking examples. For example, n ratings can be converted into n(n-1)/2 pairwise ranking.
> > > >
> > > > **A:** For a given instruction, there are 5 candidate responses from the SFT model (Alpaca-7B). As you suggested, 5 ratings on the individual responses are converted to the 10 pairwise rankings. In Table 3, we mention that we have 24.6K ratings data which is equivalent to 48K pairwise rankings. Here, 57.4% of the comparisons achieve equal rating => 27.4K comparisons.
> > > >
> > > >
> > > > **Q:** Do you think that the calibration changes of rating during the annotation progress will also happen in the ranking protocol? Or will the ranking be more stable than the rating (because there are only two options instead of 7 in the rating)? Will this difference affect the feedback quality?
> > > >
> > > > **A:** Yes, we believe that the calibration changes will also happen in the ranking protocol. However, it is hard to pinpoint such calibration changes in either of the feedback protocols. One would require conducting more controlled psychological experiments, which are out of the scope of our work but very relevant for deep understanding of feedback acquisition. In the future, it would be interesting to observe the effect of the number of options on the feedback quality and consistency behaviors. We thank the reviewer for such fine-grained feedback and comments.

---

> > > > > ### Comment · Reviewer_ofRF · 2023-11-22
> > > > > **Thanks for the further discussion**
> > > > >
> > > > > I have no more concern now.

---

### Official Review · Reviewer_gmKJ · 2023-11-01

**Soundness:** 3 good
**Presentation:** 3 good
**Contribution:** 3 good
**Rating:** 6
**Confidence:** 4

**Summary:**

This work provides an intriguing analysis of the issue of the feedback inconsistency problem, where human or model-generated evaluations can be inconsistent across different evaluation protocols (e.g., output A is better than B in pairwise evaluation while if the rating of A is lower than the rating of B). They collect 6k human feedback data under the ratings and rankings protocols, as well as model-generated feedback. They found that inconsistencies are prevalent both in human and AI evaluations. They further conduct quantitative and qualitative analyses to understand the potential factors of these inconsistencies. While such preference or rating data is essential for recent RLHF approaches as well as evaluation for open-ended generations, it is still unclear whether such data is reliable or what kind of factors affect the overall rating. This work provides an interesting analysis of this important area, and sheds light on several underexplored issues. I have several questions and concerns (e.g., inconsistencies from prior findings, validity of the final experimental), overall it has positive and good scientific contributions to ICLR.

**Strengths:**

- This paper provides an in-depth analysis of feedback acquisition by humans and AI, based on 6,000 human ranking and rating annotations for model responses, as well as model predictions.
- Their analysis reveals the prevalence of feedback inconsistency issues and also provides an in-depth analysis of why it arises.
- They also found that which type of feedback data a model is trained on has strong effects on the evaluation.

**Weaknesses:**

I found this paper quite interesting, but the paper reports a set of many different findings, and sometimes their findings are inconsistent with the previous work. Having more detailed discussions on key findings can make this paper stronger (detailed below). I am also not fully convinced by the results of Section 5. Below, I detailed those points.

**Lack of detailed discussions on findings and detailed annotation setup**

Particularily Section 3 and 4 report various interesting phenomena, but some of them lack detailed explanations. For instance,

- Section 3.1 length distributions: despite multiple papers reporting length of responses has a positive correlation with rating, the authors claim there's no difference ("we find that there is no discernible difference between the average length and average number of unique tokens of the preferred and unpreferred response in the rankings feedback collected from the humans and AI."). I wonder if the authors have any insights into this.
- Section 4.2 Qualitative analysis: the authors say that they sampled a few inconsistent instances and asked annotators to provide explanations for the inconsistencies. I think this is an important and inconsistent analysis and the annotation protocols should be precisely documented. If the claim "the differences in the preferences of the humans while annotating for different feedback protocols played a significant role in their decision making" is based on 2-3 instances, the claim may not be fully satisfied.

**The findings of "Alignment and Evaluation" section**

I think the findings of Best-of-n policies outperform SFT have been already reported in prior work, and evaluation inconsistency is somewhat predictable given the discussion of inconsistencies. While the second part (inconsistencies) can be novel, I am confused about the descriptions of the results. To my understanding, the finding is if we use a ranking model for Best-of-n ranking we can get higher rates when the same ranking protocol is used during evaluations. For me, it's not really surprising as the reward model is trained on the feedback data and reranks n responses at inference time, so if a model is trained on pairwise feedback data it learns to choose the response preferred in the pairwise setup. It'd be interesting if you could use the feedback data during training (e.g., PPO) and see if the trends remain as well. Yet, I am overall confused with the descriptions in these paragraphs and feel free to correct me if

**Questions:**

- Could you prvovide the details of human annotation process of Qualitative analysis?

---

> ### Author Response · Authors · 2023-11-18
> **Response to Reviewer gmKJ**
>
> We are excited to find that the reviewer finds our work (a) interesting in an important area, (b) sheds light on unexplored issues,  (c) and with positive and good scientific contributions to the ICLR community. Here, we address your comments in detail.
>
> **Q:** Section 3.1 length distributions
>
> **A:**
> - We thank the reviewer for their pertinent question. In Section 3.1, we mention that the max length of the Alpaca generated response is 128. This avoids humans to rank very large responses (>128 tokens).  In Table 6 and 7, we show that the human and AI is indeed not biased towards longer responses or towards the number of unique words in our setup. However, the results might differ with the changes in the feedback collection setup. We further identify the distinctions between our setup and the previous ones to further understand such differences.
> - [1] mentioned verbosity bias examined verbosity bias when using LLMs to collect feedback data by constructing a repetitive list attack in which responses with a list were prepended with a paraphrased version of the list (generated by GPT-4) which contains no new information. For example, model A has two pointer responses and model B has the same two pointers + their paraphrased versions (which does not add any new information), the annotators prefer model B response. We believe that this setting largely differs from our setting since we consider two different responses with no further intervention such as a repetition attack. Furthermore, [1] conducts their experiment on just 23 model responses [3] whereas we perform 2K responses. We will add this discussion in our updated paper.
> - Similarly, [2] talks about verbosity bias in preference labeling from humans and AI for creative writing tasks. In our work, we consider a broad range of instructions sourced from 3 datasets as discussed in Table 1. In addition, they conduct their experiments on just 300 responses whereas we perform our analysis with 2K responses. We will also include this discussion in our paper.
>
> [1] LLM-as-judge: https://arxiv.org/pdf/2306.05685.pdf \
> [2] Verbosity Bias: https://arxiv.org/pdf/2310.10076.pdf \
> [3] Data: https://github.com/lm-sys/FastChat/blob/main/fastchat/llm_judge/data/mt_bench/reference_answer/gpt-4.jsonl
>
>
> **Q:** Section 4.2: Qualitative Analysis
>
> **A:**
>
> We thank the reviewer for their insightful question. We agree that it is important to understand the cause of inconsistencies quantitatively and qualitatively. Here, we describe our qualitative analysis for more clarity:
>
> 1. In Section 4.2, we show that the feedback data collected from humans and AI is highly inconsistent and analyze their behaviours.
> 2. Further, we quantify the closeness with the model responses from the lens of rating and ranking feedback protocol. Overall, this analysis suggested that the perceived closeness between the quality of the responses leads to more inconsistent feedback.
> 3. To investigate this further, we conduct another small scale experiment. Specifically, we sample a few consistent and inconsistent responses, and then ask the annotators to provide their rating/ranking feedback along with their natural language explanation. We agree with the reviewer that such experiments should be better documented. As a result, we will include the human annotation interface in the updated paper. We believe that our small scale experimental setup is a good design under a finite budget for a qualitative study.
> 4. Post-feedback collection, we study the human responses and natural language explanations for both inconsistent (Table 9 and 10) and consistent responses (Table 11 and 12).
> 5. We notice that the natural language explanations from humans do explain their choice well. However, the quality aspects that they focus on changes with the feedback protocol.
> 6. Specifically we look at the inconsistent feedback example in Table 9. In the ranking setup, the annotators perceived Response 2 as more clever (“making a riddle..”) or correct (“makes more sense..”) over Response 1. Whereas, in the rating setup, the Response 1 gets higher average score as the annotators focussed on the answer’s faithfulness to the task instruction (“Response 2 does not meet the task structure requirements”..)
> 7. To provide more evidence, we will update the revised paper with more qualitative examples in the Appendix.

---

> > ### Author Response · Authors · 2023-11-18
> > **Response to Reviewer gmKJ**
> >
> > **Q:** The findings of "Alignment and Evaluation" section
> >
> > **A:**
> >
> > We thank the reviewer for raising their concern. We describe our process here for more clarity:
> >
> > 1. In Section 4, we show that the feedback protocols (rating/ranking) do not agree with each other i.e., 60% disagreements between the humans and AI as annotators.
> > 2. This begs the question: how does the feedback inconsistency affect our aligned LLMs and their evaluation.
> > 3. To understand this end, we train reward models on the rating and ranking data independently.
> > 4. We prompt the base Alpaca model to generate 64 responses for the 553 unseen instructions taken from AlpacaEval [1] dataset.
> > 5. We re-rank the 64 responses using the rating and relative reward model independent of each other, and choose the best response.
> > 6. Finally, we show the responses from this best-of-n strategy and a reference model (davinci-003) to the annotators.
> > 7. We ask a different set of human annotators to provide their rating and ranking feedback on the responses. In addition, we also ask ChatGPT as an annotator to provide their rating and ranking feedback on the responses. We clarify that the rating feedback is collected independently for each response whereas the ranking feedback is a pairwise evaluation.
> > 8. Finally, we calculate the fraction of times where the response from best-of-n strategy using rating/ranking reward model is preferred over the davinci-003 response.
> > 9. For the ranking evaluation, we compare if the annotators chose responses from best-of-n or davinci-003. For the rating evaluation, we compare the ratings given to the responses from best-of-n and davinci-003, and compare them. This gives us the win-rates for rating and ranking strategies using human and AI annotators.
> > 10. In Figure 3, we show that the feedback protocol chosen during evaluation prefers the responses from the best-of-n policy that utilizes the same feedback protocol. That is, ranking evaluation prefers the responses from the ranking reward model and vice-versa.
> > 11. Takeaway: Preference acquisition strategy biases evaluation.
> > 12. Our work thus highlights that the impact of different design choices can be drastic for the LLM alignment pipeline.
> >
> > We believe that our observation is novel and has not been reported in the prior work. We will be happy to provide further clarifications if there are further concerns.
> > [1] AlpacaEval: https://github.com/tatsu-lab/alpaca_eval

---

> > > ### Author Response · Authors · 2023-11-18
> > > **Response to Reviewer gmkJ**
> > >
> > > **Q:** Use the feedback data during training (e.g., PPO) and see if the trends remain as well.
> > >
> > > **A:**
> > >
> > > We thank the reviewer for their thoughtful question. Firstly, we highlight that the best-of-n (rejection sampling) is also a common method in RLHF literature for aligning LLMs. As mentioned in Section 2.3, we choose rejection sampling as due to simplicity, simplicity, and robust performance. For instance, [1] finds that the win-rate of the Best-of-n policy outperforms the base model by 10% absolute points on AlpacaEval.  We do not perform PPO since it requires atleast 8 80GB GPUs according to the AlpacaFarm implementation, which is outside our academic budget. We attempted to train PPO at our compute scale by reducing the effective batch sizes and using LoRA models but it did not run stably. Following the reviewer’s feedback, we performed an additional experiment with rejection sampling finetuning (RSFT) used in three highly cited LLM alignment papers, including LLaMA 2 [2,3,4].
> > > Specifically, our setup is as follows:
> > >
> > > 1. We prompt Alpaca-7B with 5K instructions from Alpaca-52K data.
> > > 2. We generate 64 responses for every instruction.
> > > 3. We use our rating and ranking reward model to select the best response from the 64 responses.
> > > 4. We finetune two Alpaca-7B models with the 5K instruction-responses data.
> > > (a) One where the responses are chosen from the rating reward model.
> > > (b) Second where the responses are chosen from the ranking reward model.
> > > 5. Post finetuning, we sample a single response from the finetuned Alpaca-7B with 553 evaluation instructions
> > > 6. We calculate the win-rate against DaVinci003 using the rating and ranking protocol using ChatGPT.
> > >
> > > Here are the results for this experiment where baseline is the base Alpaca-7B model:
> > > | Win-rate against Davinci-003 | Baseline | RSFT (Rating) | RSFT (Ranking) |
> > > |------------------------------|----------|---------------|----------------|
> > > | Ranking Evaluation           | 36.9%     | 42.0%          | **43.3%**           |
> > > | Rating Evaluation            | 41.9%    | **44.0%**         | 43.0%           |
> > >
> > > We find that the evaluation inconsistency persists under this alignment algorithm too. It indicates that the choice of feedback protocol for the evaluation favors the same feedback protocol used for training the reward models and subsequently finetuning the base LLM. We will add this result in the updated paper. We hope that this experiment answers the reviewer’s comment on the generalizability of the evaluation protocol to other algorithms.
> > >
> > > [1] AlpacaFarm: https://arxiv.org/abs/2305.14387 \
> > > [2] RRHF: https://arxiv.org/abs/2304.05302 \
> > > [3] LLaMA2: https://arxiv.org/pdf/2307.09288.pdf \
> > > [4] Constitutional AI: https://arxiv.org/pdf/2212.08073.pdf \
> > > [5] AlpacaFarm Code: https://github.com/tatsu-lab/alpaca_farm#running-reference-methods

---

> ### Author Response · Authors · 2023-11-20
> **Rebuttal Reminder**
>
> Thanks again for your insightful feedback on our work! We've carefully worked to address your comments/questions and would like to note that the end of the discussion phase is coming soon. Are there any further questions or concerns we should discuss?

---

> ### Comment · Reviewer_gmKJ · 2023-11-20
> **Thanks for your response.**
>
> Dear authors,
>
> Thank you so much for your response. I appreciate your detailed response, but I think the analyses could have been improved, or the descriptions are still not clear to me.
>
> Re: length bias
>
> Regarding the length issue, now I think the different finding might be coming from a much shorter maximum length---the max length of 128 seems to be really short for instruction-following queries. While I am not super familiar with [1], [2], [3] and their evaluation setups might be different, several papers analyzing the relationship between human/model preferences and length on the setup similar to this work e.g., Singhal et al. (2023) or Wang et al. (2023) set maximum length much higher. In my opinion, if we want to draw some conclusions on the length bias in human/model evaluations, we should set the maximum length longer.
>
> Also I am confused by the sentece.
>
> This avoids humans to rank very large responses (>128 tokens).
>
> Do you mean that you set the maximum tokens to be 128 as it's going to be expensive if we ask human to rank long sequences?
>
> Re: qualitative analysis
>
> Could you clarify the exact number you sampled, the overall statistics of qualitatively analyzed data, and representative trends? For instance, as in Xu et al. (2023), you can meta-analyze the human explanations to understand what common cases are.
>
> we sample a few consistent and inconsistent responses, and then ask the annotators to provide their rating/ranking feedback along with their natural language explanation. We agree with the reviewer that such experiments should be better documented. As a result, we will include the human annotation interface in the updated paper. We believe that our small scale experimental setup is a good design under a finite budget for a qualitative study.
>
> I agree that it is difficult to scale up qualitative analysis, but due to the subjective nature of qualitative analysis, qualitative analysis should be carefully designed and a paper should provide clear details of the experimental designs to make it reproducible. For me, it's hard to draw conclusions without knowing the exact details and stats of this analysis.
>
>
> Overall, I think the initial motivation is interesting and I support the paper, while I felt the analysis could have been improved, and the findings may not be fully applicable to more general settings (e.g., larger LMs, longer sequences, different way of RLHF, such as PPO). Therefore, I keep my score (6).

---

> > ### Author Response · Authors · 2023-11-20
> > **Response to Reviewer**
> >
> > We thank the reviewer for their insightful follow up comments.
> >
> > **Q:** the max length of 128 seems to be really short for instruction-following queries
> >
> > **A:**
> > We analyze the Alpaca data, a popular instruction-following dataset widely used in the community. We find that the average length of 52K responses in the Alpaca data is 45 tokens. Thus, the max length of our sampling method from the model is large enough.
> > As discussed in the paper, we clarify that our setup is not biased towards the response lengths. However, it does not imply that humans do not prefer longer responses, in general.
> >
> > **Q:** This avoids humans to rank very large responses (>128 tokens).
> >
> > **A:**  [1,2,3] aim to study this phenomenon when they show two responses with very different response lengths, whereas our setup usually shows the responses of similar lengths (<128 tokens). In terms of the financial aspect, the price of AI/human annotation does get affected with the length of the responses. For example, ChatGPT gets paid for every input token. Humans need to be paid per hour – if the responses are very long then the number of annotations per hour will reduce, hence, leading to an increase in the total cost of the experimentation. We will clarify this aspect in the revised paper.
> >
> > **Q:** Qualitative Analysis
> >
> > **A:**
> > - We clarify that this experiment is only performed with the human annotators. The group of annotators that provide rating/ranking with explanations is mutually exclusive.
> > - In total, we consider a subset of 50 instructions and 100 responses (2 responses per instruction) from our dataset. In addition, we assign 3 annotators per rating/ranking instance.
> > - The humans are provided the same annotation guidelines as before but they are asked to provide an explanation for their annotation too. We provide them 2 solved examples of how the explanations could look like.
> > - We intended to keep the human explanations free from any biases from the authors, hence, we instructed the humans to provide their explanation based on their perception of the response quality. We will add the UI screenshot of the human annotator interface for rating and ranking qualitative assessment in the revised paper.
> > - We present a few examples of the human explanations in Table 7-10 of the paper. We will add a couple of more examples in the revised version of the paper.
> >
> > We sincerely thank the reviewer for their comments and suggestions. Please feel free to ask more questions.

---

### Official Review · Reviewer_7SsY · 2023-11-06

**Soundness:** 3 good
**Presentation:** 3 good
**Contribution:** 3 good
**Rating:** 8
**Confidence:** 4

**Summary:**

This paper studied the two kinds of protocols used for collecting preference data: ratings and rankings. The authors found an inconsistency problem where in the preferences inferred from ratings and rankings significantly disagree 60% for both human and AI annotators. Their subsequent analysis identifies various facets of annotator biases that explain this phenomena such as human annotators would rate denser responses higher while preferring accuracy during pairwise judgments, for a particular comparison instance. Finally, they also found that the choice of the feedback protocol has a sharp influence on the evaluation of the aligned LLMs in the form of evaluation inconsistency. This highlights a challenge in designing robust evaluation protocols that mirror real-world performance.

**Strengths:**

- This paper studied an interesting question: the significance of Ratings versus Rankings in collecting and evaluating preference feedback. To me, this represents an essential and timely investigation.
- The findings in the paper could shed light on this question and stimulate further research.
- The paper is well structured and written. The experiments are extensive and carefully designed.

**Weaknesses:**

I liked this paper a lot but I do have several comments:

- In Table 2, the authors showed that when converting the rating to ranking, there is a huge disagreement between the two different protocols. One flaw is that a proper baseline is missing here. For example, if you annotate the same examples using another batch of human or sample from the same model (GPT-3.5-Turbo) using different temperature, what would be the disagreement? I would be curious to see such a baseline as it also correlates with the main conclusion of this paper.

- This is not necessarily a weakness but given what observed in the paper, it would be great to see if the authors could take one step further to verify the root cause of the difference and try potential ways to combine the advantages of the two annotation protocols. To me, Figure 3 shows that the ranking-based protocol is more effective and robust (it yield better win rate with ranking-based evaluation protocol and smaller gap when changing the evaluation protocol). This makes sense, as in the rating protocol, the model/human only see each individual answer thus the rating is likely to be less calibrated. Also, seen from the examples in section H, the ranking protocol is likely to introduce some systematic biases. Here is my proposal: can we try the third protocol that instruct the model/human to give both ratings and rankings for either a pair of responses or a list of responses. My guess is that rating annotated in such a way will be better calibrated and the rankings will less likely to be biased.

**Questions:**

see weaknesses

---

> ### Author Response · Authors · 2023-11-19
> **Response to Reviewer 7SsY**
>
> We are excited to find that the reviewer really liked our work and found it (a) essential and timely, (b) well structured and written, and (c) extensive and carefully designed.
>
> **Q:** Annotate the same examples using another batch of human or sample from the same model (GPT-3.5-Turbo) using different temperature, what would be the disagreement?
>
> **A:** We thank the reviewer for their insightful question. Firstly, we provide further clarifications on collecting the feedback data from the humans.
>
> 1. During the feedback acquisition, we create two exclusive groups of humans – one that provides the rating feedback and ranking feedback.
> 2. For the rating protocol, four humans rate an individual response for a given instruction.
> 3. Similarly, four humans rank a pair of responses for a given instruction under the ranking protocol.
> 4. As described in the Agreement Analysis (3.1), we consider three annotations as the predicted human preferences and the fourth annotation as the human gold label.
> 5. Table 2(b) indicates consistency analysis of the predicted human preferences i.e., randomly selecting 3 annotators out of 4 annotators.
> 6. To provide further analysis as requested by the reviewer, we calculate the consistency evaluation for a different batch of randomly selected 3 annotators four times.
> 7. We find that the standard deviation of the inconsistency analysis for four splits of the humans evaluators is **1.6%**. This indicates that the inconsistency analysis is robust to the batch of humans used for annotation.
>
> To provide more evidence for the same model using a different temperatures – 0, 0.2 and 0.5 for 1000 comparisons. We find that the standard deviation in the disagreements between these temperatures is **2.7%**.  This indicates that the inconsistency analysis is also robust to the different model temperatures used for generating feedback data. We will add these statistics in our updated paper.
>
> **Q: Root cause of the difference**
>
> **A:**
>
> We agree that the root cause of the difference between the feedback protocols is quite important. Towards this, we highlight our quantitative and qualitative findings in the Section 4.2:
>
> 1. We show that the feedback data collected from humans and AI is highly inconsistent and analyze their behaviours.
> 2. Further, we  quantify the closeness with the model responses from the lens of rating and ranking feedback protocol. Overall, this analysis suggested that the perceived closeness between the quality of the responses leads to more inconsistent feedback.
> 3. To investigate this further, we conduct another small scale experiment. Specifically, we sample a few consistent and inconsistent responses, and then ask the annotators to provide their rating/ranking feedback along with their natural language explanation. We agree with the reviewer that such experiments should be better documented. As a result, we will include the human annotation interface in the updated paper. We believe that our small scale experimental setup is a good design under a finite budget for a qualitative study.
> 4. Post-feedback collection, we study the human responses and natural language explanations for both inconsistent (Table 9 and 10) and consistent responses (Table 11 and 12).
> 5. We notice that the natural language explanations from humans do explain their choice well. However, the quality aspects that they focus on changes with the feedback protocol.
> 6. Specifically we look at the inconsistent feedback example in Table 9. In the ranking setup, the annotators perceived Response 2 as more clever (“making a riddle..”) or correct (“makes more sense..”) over Response 1. Whereas, in the rating setup, the Response 1 gets higher average score as the annotators focussed on the answer’s faithfulness to the task instruction (“Response 2 does not meet the task structure requirements”..)
> 7. To provide more evidence, we will update the revised paper with more qualitative examples in the Appendix.

---

> > ### Author Response · Authors · 2023-11-19
> > **Response to Reviewer 7SsY**
> >
> > **Q: This makes sense, as in the rating protocol, the model/human only sees each individual answer thus the rating is likely to be less calibrated.**
> >
> > - We clarify that we collect the rating data on a scale of 1-7 from AI and humans where {1=Very poor, 2=Poor, 3=Below Average,4=Average,5=Above Average,6=Good,7=Excellent}.
> > - We provide an annotation guideline in the prompt to the AI explaining the various quality dimensions along which the rating score decision must be based along with two in-context examples – one for low score scenario and second for the high score scenario. In addition, we perform rating data collection for each instance in a new session so that the LLM maintains its same calibration/understanding for the rating scores throughout the annotation process. We will add the prompt to the revised paper.
> > - We provide the same annotation guidelines as AI to humans. Differently from AI, the humans can (re)calibrate themselves as the annotation progresses. As we ask four workers to annotate the same instance, the small differences caused by the worker’s calibration will be averaged out. Specifically, the feedback data will not be biased towards a specific annotator's calibration, instead reflect the average belief about the response quality under the rating/ranking feedback protocol. We will add the UI screenshot of the human annotator interface in the revised paper.
> >
> >
> > **Q: Third protocol with a mix of rating and ranking flavors**
> >
> > Following up on the reviewer’s suggestion on mixing rating and ranking protocols, we acquired feedback data using a third protocol i.e., pairwise rating.
> >
> > **Pairwise Rating:** Given a pair of responses, provide a rating score [1-7] to both of them.
> > 1. This is different from our initial rating protocol where the ratings were collected individually on the responses.
> > 2. We convert this newly collected pairwise rating response to a ranking form by comparing the scores given the pair of responses. For example, if Response 1 got 7 and Response 2 got 5, we consider Response 1 > Response 2.
> > 3. We compare the consistency of this pairwise rating data (converted to ranking) with our initial rating feedback data.
> > 4. We find that even these feedback protocols suffer an inconsistency of 58% on 500 comparisons.
> > 5. It also highlights that the rating collected in a pairwise manner (akin to ranking format) is not calibrated with the rating collected for the responses individually.
> > 6. This can be attributed to the fact that the annotators focus on the different response quality attributes when providing pairwise feedback and independent feedback.
> >
> > We thank the reviewer for suggesting such novel experimentation. We will add the results in the updated paper.

---

> ### Author Response · Authors · 2023-11-20
> **Rebuttal Reminder**
>
> Thanks again for your insightful feedback on our work! We've carefully worked to address your comments/questions and would like to note that the end of the discussion phase is coming soon. Are there any further questions or concerns we should discuss?

---

> ### Author Response · Authors · 2023-11-21
> **Update on the revised paper**
>
> Hi,
>
> We wanted to highlight that we have uploaded the revised paper with the new results and setups. Addressing your suggestions, we have added Section E (variation with annotators), Section I (AI prompt), Section J (human UI screenshot), and Section F (pairwise rating), Section P (more qualitative examples).
>
> Please do let us know if there are any further questions. If all your questions have been resolved, we would be grateful if you would consider raising your score.

---

### Author Response · Authors · 2023-11-21
**Message to AC (Summary of all Changes)**

Dear AC,

Thank you for handling our paper. We read the reviewer’s comments carefully, and have addressed most of the major comments, including the follow-up ones. Based on the suggestions, we have added new experiments and findings in the revised paper that are highlighted in blue. (https://openreview.net/pdf?id=dKl6lMwbCy)

We highlight that the reviewers find our (1) work is essential, well-structured, and extensive (7SsY), (2) it explores unexplored issues and contributes scientifically to the ICLR community (gmKJ), and (3) it is important for understanding feedback protocols and their applicability to both humans and AI (KCah).

We briefly summarize the reviewer’s concerns and our response:

1. Reviewer gmKJ and ofRF raised the concerns about the applicability of our findings to alignment methods beyond Best-of-n policy. To that end, we ran a new experiment with rejection sampling finetuning (RSFT) which is used by popular LLM alignment papers. We find that the evaluation inconsistency is still present in the new setup, hence, adding further credibility to our experiments. Section M in the revised paper presents the setup with results.
2. Reviewer 7SsY suggested performing an experiment with a mix of rating and ranking protocols. To this end, we acquire ratings on a pair of responses from the AI annotator. We find that collecting rating feedback on individual responses and pairwise responses are also inconsistent 58% of the time. It highlights that the annotators focus on the different response quality attributes when providing pairwise feedback and independent feedback. Section F of the revised paper presents the setup results.
3. Reviewer KCah questioned the choice of models for reward modeling. To this end, we performed an additional experiment with RoBERTA-large and observed evaluation inconsistency. Appendix N of the revised paper mentions these results.
4. Reviewer ofRF wondered whether the evaluation inconsistency is due to the differences in the reward model objectives. To this end, we convert the ratings data into the ranking format and show that the evaluation inconsistency is still prevalent. Section O of the revised paper mentions these results.
5. Reviewer 7SsY and KCah enquired about the changes in the consistency analysis with the variation in the annotators. To this end, we perform feedback acquisition with different AI models and show that the inconsistencies exist for all the models. Section E in the revised paper mentions these results.
6. Reviewer 7SsY, gmKJ, orRF and KCah comment on the feedback (and explanation) acquisition protocols. To this end, we have added the AI annotator prompts in Section I and Human UI screenshot in Section J of the revised paper. We also add more additional qualitative examples in Section P.
7. Reviewer ofRF requested the feedback, and we have uploaded the data as a part of the supplementary material.

---

### Meta-Review · Area_Chair_DoYS · 2023-12-24

**Metareview:**

This paper looks at the problem of inconsistency in human- and AI-annotations for two preference methods, ranking and rating. The  inconsistencies are analyzed, and reward models trained. The research question is interesting and timely, and the research done quite well. There were some concerns raised in the reviews about the lack of discussions, lack of clarity, but these were adequately addressed in the rebuttal and the revised paper. This paper makes significant contributions in the topic of LLM training based on preferences.

**Justification For Why Not Higher Score:**

This is a nice paper in an interesting topic, but the research question and the results are not as broad in scope.

**Justification For Why Not Lower Score:**

This paper makes nice, important contributions to inconsistencies in preferences for training LMs.

---

### Decision · Program_Chairs · 2024-01-16

Accept (poster)